# Balancing *mcr-1* expression and bacterial survival is a delicate equilibrium between essential cellular defence mechanisms

Qiue Yang[1], Mei Li[1], Owen B. Spiller[1], Diego O. Andrey [1,2], Philip Hinchliffe[3], Hui Li[4], Craig MacLean [5], Pannika Niumsup[6], Lydia Powell[7], Manon Pritchard[7], Andrei Papkou[5], Yingbo Shen[8], Edward Portal[1], Kirsty Sands[1], James Spencer[3], Uttapoln Tansawai[6], David Thomas[7], Shaolin Wang[8], Yang Wang[8], Jianzhong Shen[8] & Timothy Walsh[1]

MCR-1 is a lipid A modifying enzyme that confers resistance to the antibiotic colistin. Here, we analyse the impact of MCR-1 expression on *E. coli* morphology, fitness, competitiveness, immune stimulation and virulence. Increased expression of *mcr-1* results in decreased growth rate, cell viability, competitive ability and significant degradation in cell membrane and cytoplasmic structures, compared to expression of catalytically inactive MCR-1 (E246A) or MCR-1 soluble component. Lipopolysaccharide (LPS) extracted from *mcr-1* strains induces lower production of IL-6 and TNF, when compared to control LPS. Compared to their parent strains, high-level colistin resistance mutants (HLCRMs) show reduced fitness (relative fitness is 0.41–0.78) and highly attenuated virulence in a *Galleria mellonella* infection model. Furthermore, HLCRMs are more susceptible to most antibiotics than their respective parent strains. Our results show that the bacterium is challenged to find a delicate equilibrium between expression of MCR-1-mediated colistin resistance and minimalizing toxicity and thus ensuring cell survival.

[1] Department of Medical Microbiology and Infectious Disease, Division of Infection and Immunity, Cardiff University, Cardiff CF14 4XN, UK. [2] Service of Infectious Diseases, Geneva University Hospitals and Faculty of Medicine, 1211 Geneva, Switzerland. [3] School of Cellular and Molecular Medicine, University of Bristol, Bristol BS8 1TD, UK. [4] Beijing Key Laboratory of Diagnostic and Traceability Technologies for Food Poisoning, Beijing Center for Disease Prevention and Control, Beijing 100013, China. [5] Department of Zoology, University of Oxford, South Parks Road, Oxford OX1 3PS, UK. [6] Department of Microbiology and Parasitology, Faculty of Medical Science, Naresuan University, Phitsanulok 65000, Thailand. [7] Advanced Therapies Group, Oral and Biomedical Sciences, School of Dentistry, College of Biomedical and Life Sciences, Cardiff University, Heath Park, Cardiff CF14 4XN, UK. [8] Beijing Advanced Innovation Centre for Food Nutrition and Human Health, College of Veterinary Medicine, China Agricultural University, Beijing 100193, China. Correspondence and requests for materials should be addressed to Q.Y. (email: YangQe@cardiff.ac.uk) or to T.W. (email: WalshTR@cardiff.ac.uk)

The global concern over colistin resistance has profound implications as to how we successfully manage and treat serious Gram-negative infections, particularly those caused by *Enterobacteriaceae*[1,2]. Until recently, our understanding of colistin resistance was limited to chromosomal changes such as the *pmrA/pmrB* activation of *arnBCADTEF* and *pmrE*, that collectively modifies lipopolysaccharide (LPS), the component of the Gram-negative outer membrane by the addition of 4-amino-4-deoxy-L-arabinose[3]. However, the advent and subsequent reporting of plasmid-mediated colistin resistance, *mcr-1*, in both animal and human health, has invoked commentaries announcing a return to the pre-antibiotic era. Since our lab described *mcr-1* in November 2015, it has been reported in over 40 countries. Regrettably, as the main bacterial host of *mcr-1* is *Escherichia coli*, there are also significant public health[4] and environmental issues as MCR-1-positive *E. coli* (MCRPEC) has been reported from migratory birds[4,5], flies[4], dogs[4], imported reptiles[6], rivers[7] and inevitably public water facilities[8]. The widespread reporting of *mcr-1* since the first sequence was published is unprecedented above any previous resistance mechanism[9,10]. This is, in part, also due to increased surveillance; nonetheless, its rapid global dissemination only further increases the critical need for better global biosafety containment and novel drug development.

The clinical impact of *mcr-1* is still unknown. Of the few clinical studies reporting MCRPEC from infections, most lack assessment of risk factors to determine whether MCRPEC have a different pathogenic profile to non-MCRPEC[11–14]. In China where MCRPEC is disseminated throughout the Chinese food industry and environment, it has been shown to colonise healthy volunteers and patients[15]. Whether the expression of *mcr-1* gene modulates pathogenicity in *E. coli* remains to be explored.

Those few studies that have examined the association of MCRPEC with virulent factors and pathogenicity have shown little or no difference[16]. However, these studies involved only a few isolates and applied different methods, so a direct correlation remains unclear. Notwithstanding, *mcr-1* plasmids have been reported in variety of different MLST clades only some of which fall into the B2 group of human pathogens[10,16]. Most noticeable, is that *mcr-1* is now starting to emerge in ST131, a virulent strain associated with a variety of clinical infections[10,17]. However,

*mcr-1* encodes an enzyme that catalyses the transfer of phosphoethanolamine onto a phosphate of the *N*-acetylglucosamine head group of lipid A in the bacterial outer membrane (Fig. 1)[18,19], which may modify the structure of lipid A and alter its ability to induce the innate immune response and modify the clinical pathogenicity of bacterial infections. Furthermore, colistin MICs displayed by MCRPEC are moderate (usually 2–8 mg l$^{-1}$) when compared to the level of colistin resistance (usually 8–256 mg l$^{-1}$) mediated by, for example, increased expression of *pmrA/pmrB*, inferring that the expression of *mcr-1* is tightly controlled[3,9,10,16]. This, in part, is supported by recent studies on plasmid copy number where *mcr-1* were only found on plasmids of a relatively low copy number[20,21].

Here, we investigate the impact of *mcr-1* expression on *E. coli* survival, fitness and virulence. Furthermore, we analyse the viability, virulence and high-level colistin resistance stability of MCRPEC high-level colistin-resistant mutants.

## Results

**Generic background and plasmid copy number.** The wild-type MCRPEC isolates (PN16, PN21, PN23, PN24, PN25, PN42 and PN43) chosen for this study were selected from a collection from Phitsanulok, Thailand (Supplementary Table 1) and in keeping with the known epidemiology of MCRPEC[4]. Isolates were chosen from a variety of sources: chicken meat (PN16), chicken faeces (PN21), duck faeces (PN23, PN24 and PN25) and human faeces (PN42 and PN43) (Supplementary Table 1). Isolates PN16 and PN21, possessed *mcr-1* plasmids with an IncI2 backbone and PN23, PN24, PN25 and PN42 possessed IncX4 *mcr-1* plasmids. Unusually, *mcr-1* was carried on the chromosome of isolate PN43 (Supplementary Fig. 1). The genomes of the strains were sequenced and varied considerably in the number and variety of antibiotic resistance genes. For example, apart from *mcr-1*, PN23 possessed one additional gene, *tetB*, where as PN42 carried numerous antibiotic resistance genes (Supplementary Table 1). The genetic maps showing the immediate context of *mcr-1* in the wild-type isolates are shown in Supplementary Fig. 2. The MLSTs for each MCRPEC were grouped as follows: ST2040 (PN16), new variant of ST24 (PN21), ST1211 (PN23), ST3631 (PN24), ST101 (PN25), ST744 (PN42) and ST410 (PN43), confirming that the

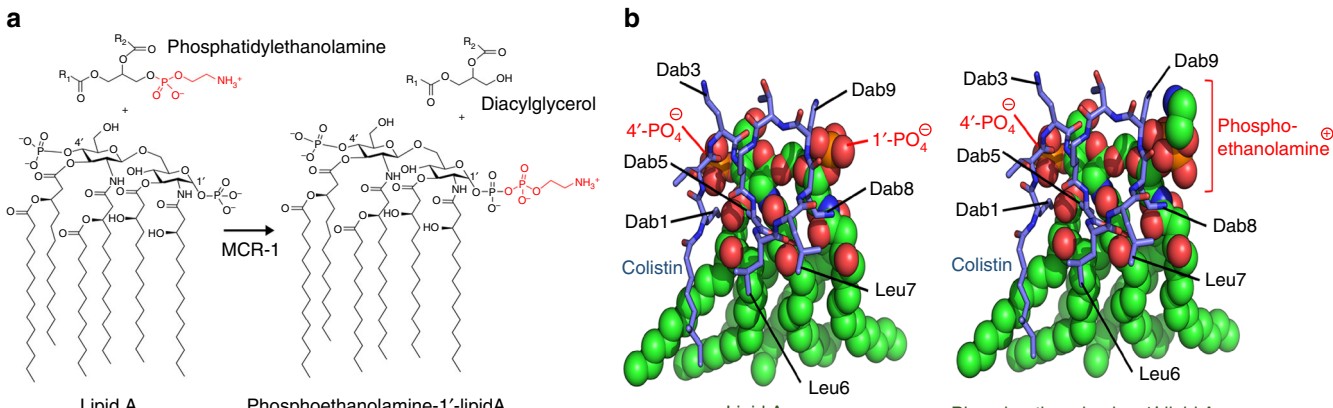

**Fig. 1** Molecular model of colistin binding to lipid A. **a** Schematic of phosphoethanolamine transfer to the 1-PO$_4$ group of hexa-acylated lipid A as catalysed by MCR-1. **b** Models of colistin (blue sticks) binding to lipid A (left) or phosphoethanolamine-lipid A (right) (spheres coloured green, red, blue and orange for C, O, N and P atoms, respectively). The model is based on the NMR and docking studies of polymyxin B binding to lipid A with lipid A coordinates from PDB 3fxi[22] and colistin coordinates adapted from the NMR structure of polymyxin B bound to lipid A[23]. The positively charged Dab colistin residues closely interact with the negatively-charged 1′ and 4′ phosphate groups of lipid A, reducing the net-negative charge of lipid A. The hydrophobic leucine residues and tail of colistin A interact with the fatty acid tails of lipid A, allowing colistin A to insert into, and disrupt, the bacterial outer membrane. **b** (right), model of colistin binding to phosphoethanolamine-lipid A indicates addition of positively charged phosphoethanolamine onto the 1′-PO$_4$ of lipid A likely interferes with the interaction of positively charged Dab8 and Dab9 side chains with the phosphate group, preventing colistin binding to the outer membrane of Gram-negative bacteria. Figure created using Pymol (https://www.pymol.org/)

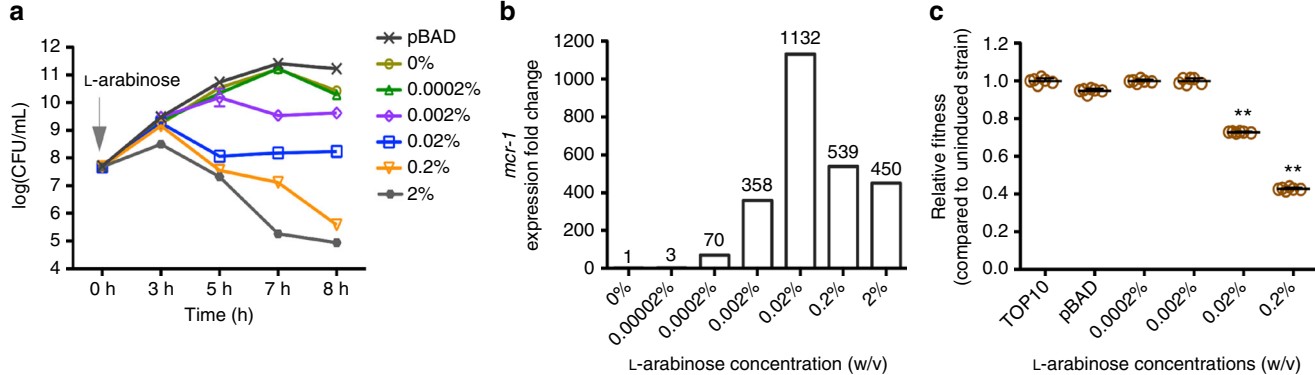

**Fig. 2** Effects of *mcr-1* overexpression on bacterial growth and fitness in vitro. **a** Overproducing *mcr-1* causes variable effects on growth rate depending on the concentrations of L-arabinose (*n* = 3). **b** The expression levels of *mcr-1* gene induced by increasing concentrations of arabinose were measured by qRT-PCR (*n* = 2). **c** Relative fitness of *mcr-1* overexpressing strain *mcr-1*/pBAD competing control strain pHT315 under increasing concentrations of L-arabinose (0.0002% vs 0%, 0.002% vs 0%, 0.02% vs 0%, 0.2% vs 0%). Error bars represent the SD (*n* = 6). The differences in fitness were tested using non-parametric Mann–Whitney test, ** indicates the *p* values is <0.05. The average relative fitness and *p* values are listed in Supplementary Table 6

MCRPEC isolates used in this study are not clonal (Supplementary Table 1). MRCPEC possessing some of these MLST groups have been reported previously[10,16]. The copy number of *mcr-1* was assessed by qPCR (quantitative PCR) using 16S ribosomal RNA as a standard reference copy number. All isolates possessed *mcr-1* at a copy number ranging between 1 (chromosomal copy of isolate PN43) and 5.6 in keeping with other recent studies[20,21] (Supplementary Table 1).

**Effects of *mcr-1* overexpression on bacterial growth and fitness.** The *mcr-1* coding region obtained from pHNSHP45 was constructed into plasmid pBAD to generate strain E. coli TOP10 *mcr-1*/pBAD (*mcr-1*/pBAD) (Supplementary Table 2, see Methods). To determine whether *mcr-1* affects bacterial growth rate and fitness, we examined growth curves, quantitative real-time PCR (qRT-PCR) and competition assays for E. coli TOP10 carrying the *mcr-1*-pBAD plasmid construct. The expression level of *mcr-1* was induced by increasing concentrations of L-arabinose and measured by qRT-PCR (Fig. 2b). Maximal *mcr-1* induction was observed at 0.02% L-arabinose, where *mcr-1* expression was approximately twofold and threefold more than the arabinose concentrations of 0.002% and 0.2%, respectively. As shown in Fig. 2a, after 8 h, the growth rates of E. coli strains induced by 0.2% and 0.02% (w/v) of L-arabinose showed approximately a 3-$\log_{10}$ unit decrease when compared to E.coli TOP10 (*mcr-1*/pBAD) without L-arabinose. As the initial inoculum was ~7.5 $\log_{10}$ (c.f.u. per ml), and this decreased to ~5 $\log_{10}$, it also suggests overexpression of *mcr-1* decreases cell populations. In vitro competition experiments were also performed to determine the effect of *mcr-1* expression on the relative fitness of E. coli (*mcr-1*/pBAD) with different *mcr-1* levels (Methods). Results are shown in Fig. 2c and indicate that increased *mcr-1* expression levels were associated with a significant fitness burden in vitro. For example, inducing high levels of *mcr-1* expression with 0.2% L-arabinose remarkably decreases relative fitness (average relative fitness 0.43, *p* = 0.0022, using non-parametric Mann–Whitney test) by >50% relative to uninduced controls (Fig. 2c and Supplementary Table 6). To eliminate these observations being singularly due to the overexpression of a random protein, our negative control, *bla*_TEM-1b_, when overexpressed showed very similar growth curves to the uninduced E. coli cells (Supplementary Fig. 4C).

**Overexpression of *mcr-1* affects cellular morphology.** As *mcr-1* encodes the transfer of phosphoethanolamine onto a phosphate of the N-acetylglucosamine head group of lipid A in the bacterial outer membrane[9]. We employed transmission electron microscopy (TEM) and study the cellular morphology of E. coli (TOP10) under induced (0.2% (w/v) L-arabinose) and non-induced conditions (Fig. 3). Both control strains, E. coli TOP10 with pBAD minus *mcr-1*, and E. coli TOP10 (*mcr-1*/pBAD) without L-arabinose induction, showed normal cellular characteristics with a multi-layered cell surface consisting of a distinct, structured outer membrane, a peptidoglycan layer in the periplasmic space, a normal cytoplasmic membrane and typically granular cytoplasm (Fig. 3a and Fig. 3b). Significant morphological changes were however, observed in E. coli TOP10 (*mcr-1*/pBAD) treated with 0.2% L-arabinose for 8 h (Fig. 3c). In particular, with the outer cell membrane cell envelope (Fig. 3f). The outer membrane region exhibited altered structural integrity, varying markedly in thickness and density, and it could not be differentiated from the cell wall or the cytoplasmic membrane (Fig. 3d–f). TEM analysis on E. coli where *mcr-1* is overexpressed shows other gross cellular changes including complete loss of the 'bacilli' morphology and absence of electron-dense material appeared as empty 'ghost' cells (Fig. 3c). By comparison, E. coli TOP10 with pBAD minus *mcr-1*, and E. coli TOP10 (*mcr-1*/pBAD) without L-arabinose induction possess homogeneous electron densities in the cytoplasm and exhibited unaltered multilayers of cell membrane (Fig. 3a, b) confirming that it is the high-level expression of *mcr-1* that has induced these gross morphological changes. In order to rule out effects caused by high concentration of L-arabinose, we examined E. coli TOP10 with pBAD alone (minus *mcr-1*) with 0.2% L-arabinose induction (Supplementary Fig. 3). Induction resulted in an intact cell wall with a well-defined inner and outer membrane, and a highly homogeneous electron density in cytoplasm were observed (Supplementary Fig. 3), further supporting that cell membrane impairment is due to the expression of *mcr-1* gene only.

**Effects of *mcr-1* overexpression on bacterial survival.** As the growth curves from Fig. 2a suggest that overexpression of *mcr-1* reduces cell counts, and given its profound effect on the E. coli cell structure and morphology, we investigated the effects of *mcr-1* on cellular viability and biofilm assembly, using LIVE/DEAD® staining with confocal laser scanning microscopy (CLSM) imaging and COMSTAT analysis in E. coli TOP10 (*mcr-1*/pBAD) biofilms with/without 0.2% L-arabinose induction (Fig. 4). The observed marked reduction of viability in bacterial cell overexpressing *mcr-1* (Fig. 4), was contrast with the minimal reduction in bacterial viability observed in E. coli TOP10 with pBAD minus *mcr-1*, and E. coli TOP10 (*mcr-1*/pBAD minus L-arabinose

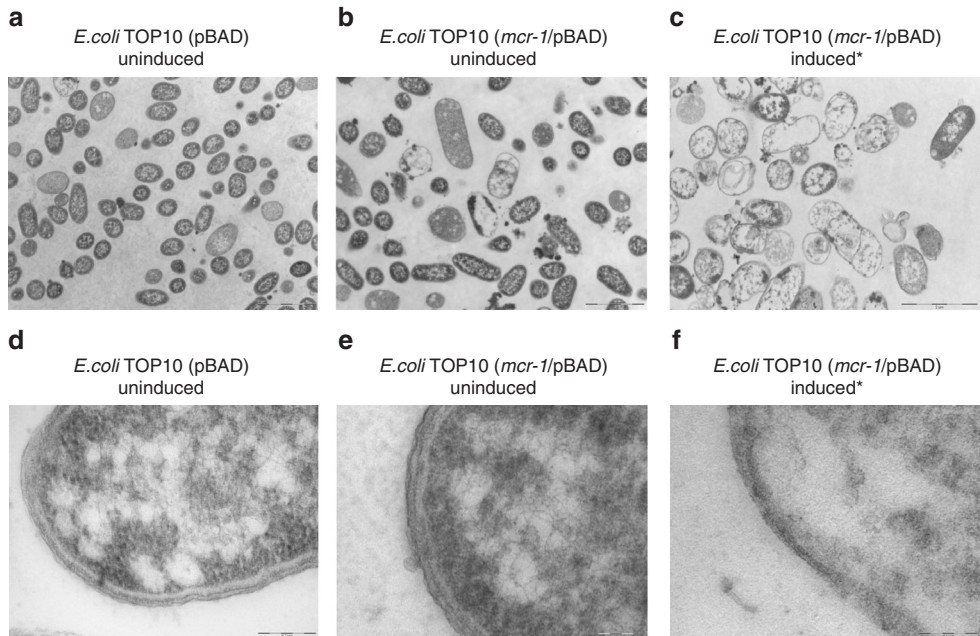

**Fig. 3** TEM micrographs of untreated and treated *E.coli*. In **a** and **b**, TEM micrographs of untreated control cells (*E. coli* TOP10 with pBAD minus *mcr-1*, and *E. coli* TOP10 (*mcr-1*/pBAD) without ʟ-arabinose induction, respectively); both cells are intact with a well-defined inner and outer membrane, and showed a highly homogeneous electron density in cytoplasm region (**d** and **e**). **c** TEM micrographs of *mcr-1* overproducing cells; the damaging outer membrane and some completely lysed cells were observed (**f**)

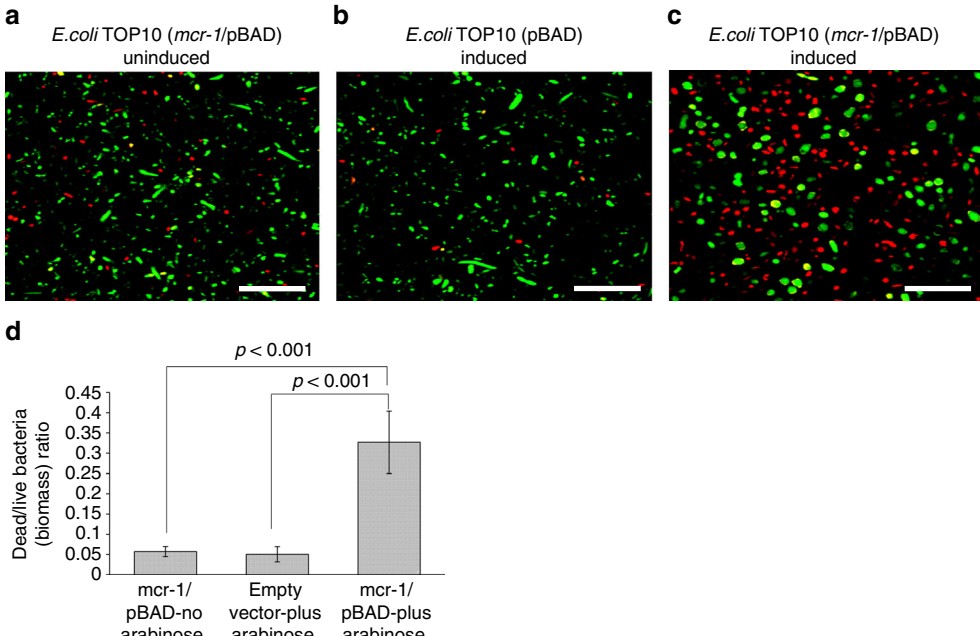

**Fig. 4** The toxic effects of *mcr-1* overexpression on cell viability. **a–c** Confocal laser scanning microscopy images of cells treated with/without ʟ-arabinose and stained with LIVE/DEAD® (*n* = 3). Live and dead cells presented green and red colour, respectively. Scale bar is 15 μm. **d** Ratio of dead to live bacteria (biomass) obtained from CLSM z-stack images through COMSTAT analysis of *E. coli* biofilms grown for 16 h in LB broth, followed by ±ʟ-arabinose (0.2% w/v; 8 h) treatment, where the biofilms were stained with LIVE/DEAD® (*n* = 4). The COMSTAT data was assessed using one-way analysis of variance (ANOVA) followed by Tukey–Kramer multiple comparisons post hoc test. Statistical significance was set at $p < 0.05$

induction) (Fig. 4a, b). COMSTAT analysis of the CLSM biofilm images revealed that relative dead/live biomass ratio (Fig. 4d) of induced *E. coli* TOP10 (*mcr-1*/pBAD) was at least six times higher than *E. coli* TOP10 with pBAD minus *mcr-1*, and *E. coli* TOP10 (*mcr-1*/pBAD) minus arabinose (0.33 vs 0.06; $p < 0.001$, using one-way analysis of variance (ANOVA)). Morphological alterations were also apparent in CLSM imaging, where the live cells (green) in Fig. 3c appeared more spherical and 'bloated', in keeping with the changes observed in the TEM studies (Fig. 3c). Overexpression of $bla_{TEM-1b}$ showed no evidence of increased cell death compared to that of *E. coli* TOP10 with pBAD minus *mcr-1*, and *E. coli* TOP10 (*mcr-1*/pBAD minus ʟ-arabinose induction

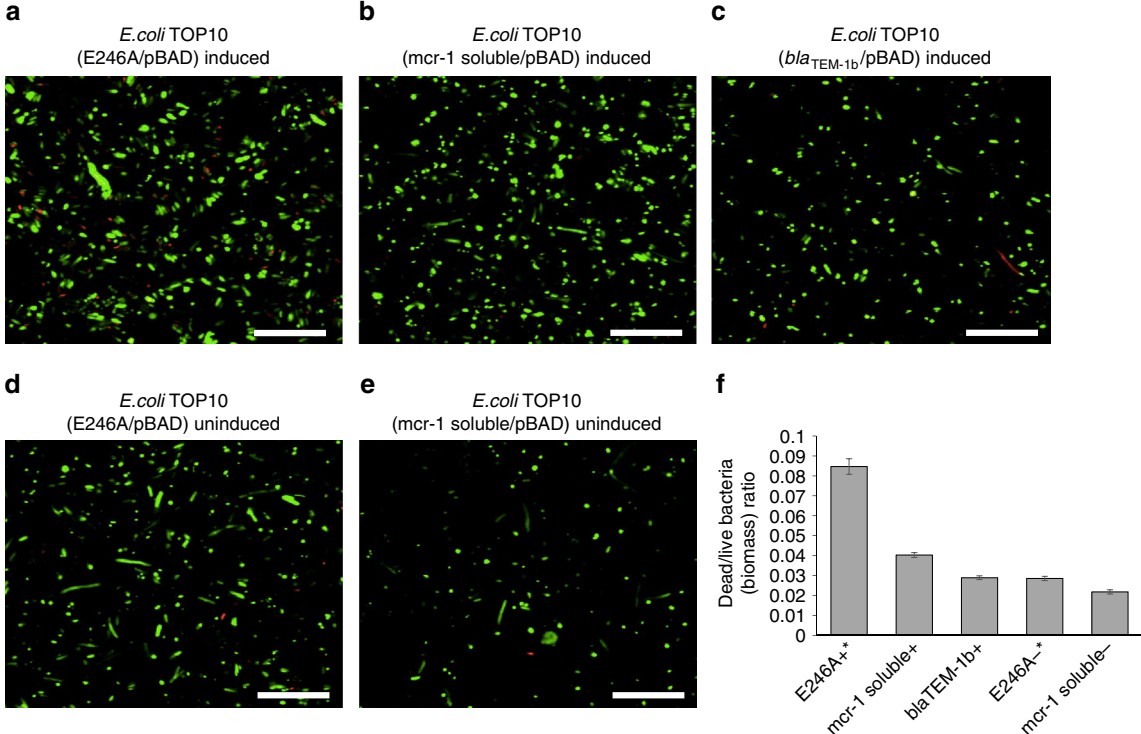

**Fig. 5** The toxic effects of *mcr-1* mutants on cell viability. **a**–**c** Confocal laser scanning microscopy images of cells treated with L-arabinose (0.2% w/v; 8 h) and stained with LIVE/DEAD® ($n = 4$). Live and dead cells presented in green and red colour, respectively. **d** and **e** Confocal laser scanning microscopy images of control cells treated without L-arabinose for 8 h and stained with LIVE/DEAD® ($n = 4$). Live and dead cells presented in green and red colour, respectively. **f** Ratio of dead to live bacteria (biomass) obtained from CLSM *z*-stack images through COMSTAT analysis ($n = 4$). The COMSTAT data was assessed using one-way analysis of variance (ANOVA) followed by Tukey–Kramer multiple comparisons post hoc test, '+' and '−' indicate bacteria treated with or without L-arabinose, respectively

(Fig. 5c)) providing further evidence that the cellular effects seen with increased expression of MCR-1 are not due to the fitness burden of protein expression.

***mcr-1*-mediated membrane-bound resistance mechanism**. Our previous study has shown that MCR-1 is a membrane-bound enzyme consisting of five hydrophobic transmembrane helixes and a soluble form located in the periplasmic domain[9]. To examine whether the transmembrane domain or MCR-1 catalytic domain (typified by aspartate at position 246) plays a role on cell integrity and bacterial fitness, MCR-1 (E246A), and a MCR-1 soluble domain (residues 219–541 and lacking the N-terminal membrane-bound region) mutant, were cloned into a pBAD-hisA plasmid. Both constructed plasmids were transferred into TOP10 cells and induced with 0.2% (w/v) L-arabinose at exponential phase. Results show that increased expression of MCR-1 (E246A) produced a toxic effect on the *E. coli* cells as evidenced by decreased growth rate (Supplementary Fig. 4), significant membrane degradation (Supplementary Fig. 5) and moderate fitness loss (Fig. 6a). In contrast, the increased expression of the MCR-1 soluble domain (219–451) did not show any marked changes in growth rate, fitness loss or membrane architecture (Fig. 5b, f, Fig. 6a, and Supplementary Figs. 4 and 5). However, the toxic effects of the MCR-1 (E246A) mutant is moderate, compared to the marked effect of MCR-1 wild-type (Figs. 2–4). For example, increased expression of the full-length *mcr-1* gene caused at least three times higher bacterial death than overproducing incomplete MCR-1 (E246A) (dead/live bacteria ratio, 0.33 vs 0.09) (Fig. 5a, f). Therefore, our data indicate that the fitness loss and the destruction of the membrane architecture is due to both the embedding of the protein in the *E. coli* outer membrane and the

phosphoethanolamine modification of LPS. Phosphoethanolamine addition on lipid A in MCRPEC has been observed by ESI-QTOF/MS (Supplementary Fig. 6), compared to that of non-MCRPEC.

***mcr-1*-mediated LPS modification reduced the stimulation of macrophage**. *mcr-1* encodes an enzyme that catalyses the transfer of phosphoethanolamine onto a phosphate of the N-acetylglucosamine head group of lipid A, and its expression affects cell morphology (Fig. 3) and survival (Fig. 4). Therefore, to investigate whether the modified LPS of MCRPEC alters the activation of human macrophage THP-1, LPS-mediated human macrophage stimulation assays were undertaken. The macrophages were activated by serial concentrations (4.5, 0.45, 0.045 and 0.0045 ng per ml) of LPS extracted from *E. coli* TOP10 with pBAD minus *mcr-1*, and *E. coli* TOP10 (*mcr-1*/pBAD plus 0.2% L-arabinose). The production of interleukin (IL)-6 and tumour necrosis factor (TNF) were assayed by using DuoSet® ELISA kit (R&D systems, UK). The concentrations of IL-6 produced by macrophages induced by unmodified LPS were consistently higher than IL-6 levels produced by MCR-1 modified LPS at 8 and 24 h (Fig. 7a, b). Additionally, TNF-alpha levels were also higher in macrophages stimulated by unmodified LPS than compared to modified LPS (Fig. 7c).

**Acquisition and stability of high-level colistin resistance mutants**. MCRPEC was first discovered on the premise that *E. coli* can rarely acquire colistin resistance by chromosomal mutations alone and that the levels mediated by *mcr-1* are moderate compared with other mechanisms[1,2,9]. Data

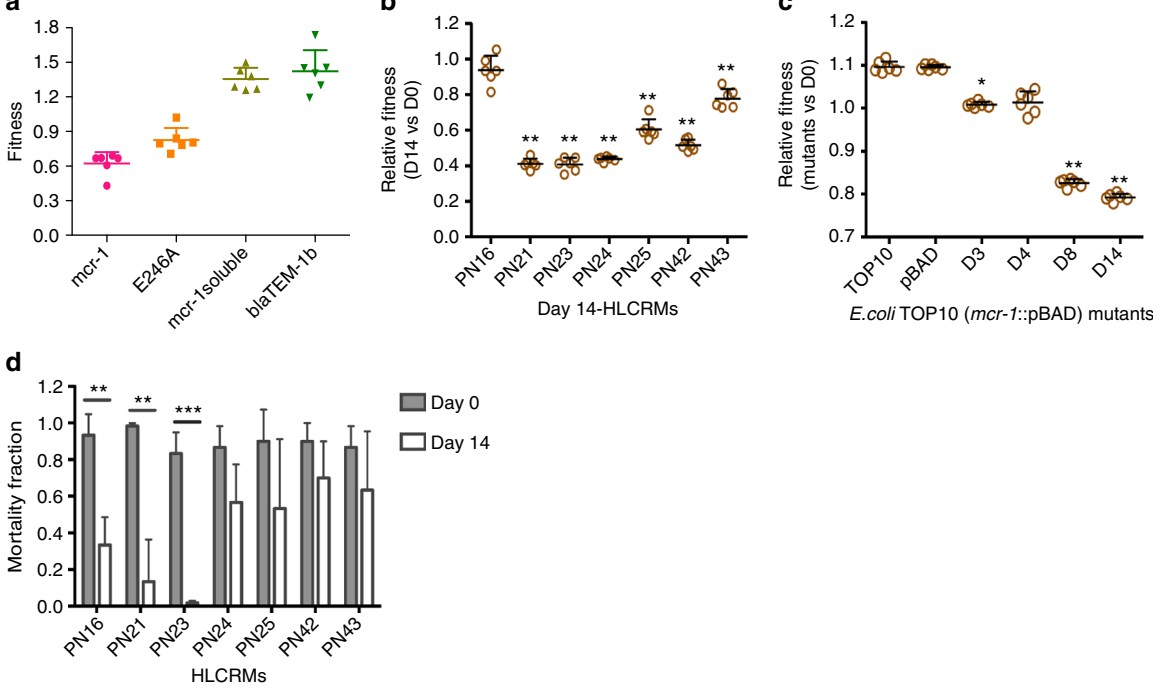

**Fig. 6** Competition assays and *G. mellonella* killing models in HLCRMs. **a** Fitness of full-length *mcr-1*, calalytically inactivated *mcr-1*(E246A), and *mcr-1* soluble domain. Fitness of blaTEM-1b as a negative control. **b** Relative fitness of wild-type *mcr-1*-positive strains and their derivatives. **c** Relative fitness of *mcr-1*/pBAD-positive *E.coli* TOP10 strains and their derivatives. In all fitness figures, error bars represent the SD ($n = 6$). The differences in fitness were tested using non-parametric Mann–Whitney test, * indicates $0.01 < p$ value $< 0.05$ and ** indicates $p$ value $< 0.01$. The average relative fitness and $p$ values are listed in Supplementary Table 6. **d** HLCRMs displays impaired virulence in the *G. mellonella* infection models. All results represent means of three independent experiments with 10 larvae per treatment. Mortality bar charts were plotted using the Kaplan–Meier method (GraphPad Software). Error bars represent the SD ($n = 3$) and $p$ value for strains PN16 ($p = 0.0056$, $t = 5.427$, d.f $= 4$), PN21 ($p = 0.0029$, $t = 6.509$, d.f $= 4$) and PN23 ($p = 0.0003$, $t = 2.183$, d.f $= 4$) were calculated by student $t$ test. ** indicates $0.001 < p$ value $< 0.01$, *** indicates $p$ value $< 0.001$

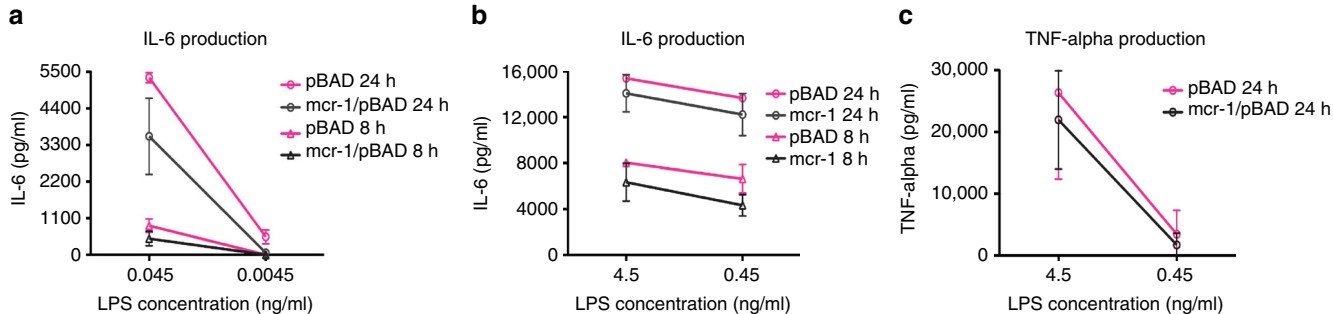

**Fig. 7** Expression of cytokines in LPS-mediated THP-1 macrophage. **a**, **b** and **c** indicated IL-6 ($n = 2$) and TNF-alpha ($n = 3$) productions in the modified LPS (grey) or normal LPS (red)-treated cell culture, respectively. Error bars represented SD

previously published by us would suggest that the level of in vivo colistin resistance mediated by MCRPEC can protect the cells compared to control strains[9]. However, the data from Figs. 2–4 would suggest that the expression of *mcr-1* is also very tightly controlled because of the fitness cost of increased *mcr-1* expression. We explored whether MCRPEC could generate high-level colistin resistance mutants (HLCRMs) over non-MCRPECs and investigated how damaging these changes would be. HLCRMs were generated from the seven wild-type MCRPEC strains described in Supplementary Table 1. These isolates possessed low-level resistance to colistin (4–8 mg l⁻¹), and HLCRMs were generated through exposure to increasing concentrations of colistin. After a 14-day serial colistin challenge, all seven wild-type MCRPEC strains exhibited an increase in colistin resistance with a four to 64-fold increase in colistin

MICs (Supplementary Fig. 7A and Supplementary Table 4). The rate of increase and final level of resistance varied considerably, with PN16 and PN23 reaching 256 mg l⁻¹; PN21, PN25 and PN43 reaching 128 mg l⁻¹; PN42 reaching 32 mg l⁻¹, yet PN24 could be increased to 16 mg l⁻¹.

We also exposed *E. coli* TOP10 (*mcr-1*/pBAD), *E. coli* TOP10 (pBAD) and *E. coli* TOP10 to the same level of passaging (Supplementary Fig. 7B and Supplementary Table 5). *E. coli* TOP10 (*mcr-1*/pBAD) (in the absence of arabinose) displayed a 64-fold (from 0.5 to 32 mg l⁻¹) increase in colistin MICs. However, both *E. coli* TOP10 (pBAD) and *E. coli* TOP10 did not show any elevation in colistin MICs remaining at 0.125 mg l⁻¹ ±1 dilution over the 14-day challenge. These data indicate that the increase in colistin resistance shown by *E. coli* TOP10 (*mcr-1*/pBAD) is singularly due to the presence of *mcr-1*. Both the expression level and copy number

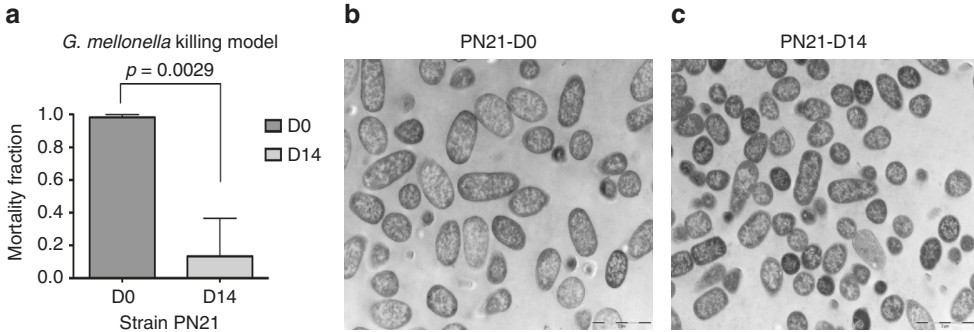

**Fig. 8** *G. mellonella* killing data and TEM micrographs of strain PN21. **a** *G.mellonella* mortality rate in PN21 parental strain (D0) and its mutant (D14). Error bars represent the SD ($n = 3$) and *p* value was calculated by Student's *t* test ($t = 6.509$, d.f = 4). **b** and **c** indicated TEM micrographs of parental strain PN21 (D0) and PN21(D14) mutant, respectively. Membranes for both strains are intact with a highly homogeneous electron density in cytoplasm region

of *mcr-1* gene were determined by qRT-PCR for all MCRPEC isolates (Supplementary Fig. 8 and Supplementary Table 1). Compared with the wild-type strains, no significant difference in plasmid copy number and *mcr-1* expression could be observed in the corresponding HLCRMs except for MRCPECs isolates PN21 and PN25, where the expression of *mcr-1* has increased by ~11-fold and threefold, respectively (Supplementary Fig. 8).

Given the fitness cost of *mcr-1* to *E. coli* and the fact that the HLCRMs are likely to mediate higher resistance in vivo and be more problematic to treat, the stability of these HLCRMs needed to be assessed for their high-level colistin resistance stability. Accordingly, the 14-day challenged HLCRMs were subsequently serially passaged in colistin-free medium to determine if the high-level colistin-resistance phenotype could be reversed. Three MCRPECs, PN24, PN25 and PN42, showed the same level of colistin resistance as their respective HLCRMs (day-14 mutants), after 14-day passages in lysogeny broth (LB) broth without colistin indicating that the high-level of colistin resistance was very stable (Supplementary Fig. 7C). However, MCRPECs PN16, PN21, PN23 and PN43 showed a marked decreased in colistin resistance during passaging in antibiotic-free medium with HLCRM PN43 being the most unstable (highest loss of colistin resistant population) followed by HLCRM derived from isolates PN23, PN16 and PN21 (Supplementary Fig. 7C).

**In vitro relative fitness of HLCRMs**. Our data examining the physiological burden of *mcr*-1 would suggest that the HLCRMs might even be defective in fitness and pathogenicity. Therefore, to compare the growth rates of the wild-type MCRPEC isolates with their respective HLCRMs, we measured growth over a serial time course of HLCRMs strains that showed different resistance levels of resistance to colistin at day 0, 3, 7, 11 and 14. Most of the HLCRMs grew up to threefold slower than their wild-type parent isolates and, in particular, those mutants recovered at day 11 and day 14 (Supplementary Fig. 9A–G). The exception being PN16 where there was no significant difference in growth rate between the parent and its corresponding HLCRM (even day-14 mutant) (Supplementary Fig. 9A). Additionally, we also examined the growth curves of HLCRMs generated from *E. coli* TOP10 (*mcr*-1/pBAD). Similarly, day-8 and day-14 mutants (both MICs of 16 and 32 mg l$^{-1}$ for colistin, respectively) showed a decreased growth compared to that of their parental strains (Supplementary Fig. 9H).

Based on our data from bacterial growth curves, we propose there is substantial fitness disadvantage to the HLCRMs when compared to their MCRPEC parent strains. To test this hypothesis, in vitro competition assays were performed and HLCRMs were directly competed against their MCRPEC parental strains. The fitness of the MRCPEC parent strains and HLCRMs were relative to the growth of the control strain that was fixed at

1.0 (Methods) (Fig. 6b). Most (6/7) of the day-14 HLCRMs had lower competitive fitness than that of their corresponding parent strains (Fig. 6b) with HLCRMs derived from PN21, PN23, PN24, PN25 and PN42 possessing a relative fitness of 0.4–0.6 and HLCRM PN43 and relative fitness rate of 0.78 (Mann–Whitney test-corrected $p = 0.0022$) (Fig. 8a and Supplementary Table 6). The PN21 and PN23 mutants, the former has 11-fold increased *mcr-1* gene expression (Supplementary Fig. 8), were associated with the highest fitness burden (~0.41, $p = 0.0022$) (Fig. 6b and Supplementary Table 6). The exception was HLCRM derived from PN16 that had a fitness rate similar to its parent strain (~0.94, $p = 0.1320$) (Fig. 6b and Supplementary Table 6). We also examined the fitness cost for *E. coli* TOP10 (*mcr-1*/pBAD), *E. coli* TOP10 (pBAD) and *E. coli* TOP10 (Fig. 6c). Day 3 and 4 HLCRMs of *E. coli* TOP10 (*mcr-1*/pBAD) had slightly less fitness (1.0 vs 1.1 of *E. coli* TOP10 (pBAD) and *E. coli* TOP10) and day 8 and 14 were less fit (0.83 vs 1.1, and 0.79 vs 1.1, respectively) (Fig. 6c). Generally, our data show a significant fitness burden with HLCRMs compared to their MCRPEC parent strains.

**Virulence reduction of MCRPEC and HLCRMs**. To determine whether the level of virulence of HLCRMs have been reduced compared to the MCRPEC parental strains, we used *Galleria mellonella* as a model of infection[24,25]. Larvae were infected with each of the MCRPEC parental strains and their respective HLCRMs, with differing bacterium inoculums in order to illicit appropriate live and dead larvae populations. The mortality of larvae was dependent on the number of bacteria injected (data not shown) and this was varied to show the greatest effect with each parent/mutant pair. As shown in Fig. 6d and Supplementary Fig. 10, a significant decrease in *G. mellonella* killing was observed in all HLCRMs, compared to their respective wild-type parent strains. In particular, HLCRMs derived from MCRPECs PN16, PN21 and PN23 gave mortality fractions of 0.33 ($p = 0.0056$, Student's *t* test), 0.13 ($p = 0.0029$) and 0.02 ($p = 0.0003$), respectively, compared to their respective parent strains that were between 0.8 and 1.0. Because the MCRPEC parental strains are highly heterogeneous with respect to their genomic background, the infection of *G. mellonella* resulted in different larvae survival rates (Supplementary Fig. 10). However, when we compared the morphological changes using TEM analysis, there was no significant change between mutants and their corresponding MRCPEC parents (Fig. 8b, c), even there was a decrease in *G. mellonella* killing ($p = 0.0029$) exhibited in PN21 (Fig. 8a).

In wild-type strains, the level of colistin resistance mediated by *mcr-1* gene is often moderate (2–8 mg l$^{-1}$). The in vitro generated HLCRMs possessed reduced fitness and virulence. We also determined that even the low level of colistin resistance mediated by *mcr-1* gene in clinical strains attenuates bacterial virulence in

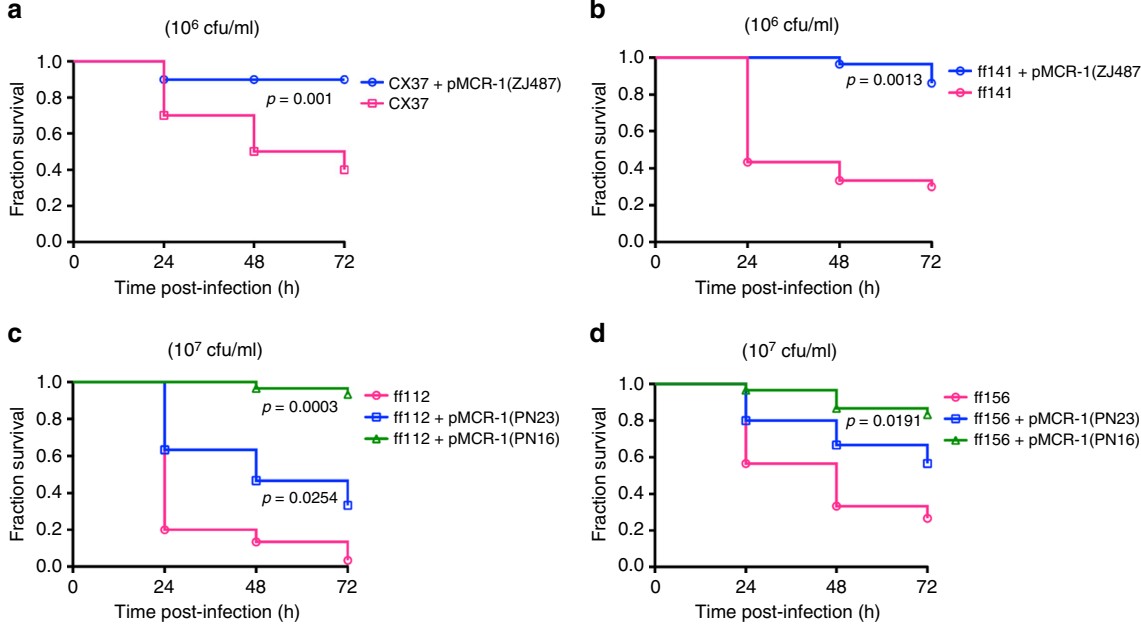

**Fig. 9** Kaplan–Meier plots showing the percent survival of *G. mellonella* over 72 h post infection with MCRPEC and non-MCRPEC human clinical strains. Survival curves were plotted using the Kaplan–Meier method (GraphPad Software). Error bars represent the SD ($n = 3$) and $p$ value for **a** ($t = 8.654$, d.f = 4), **b** ($t = 8.050$, d.f = 4), **c** ($t = 12.07$, d.f = 4 for strain ff112 with pMCR-1(PN16) and $t = 3.479$, d.f = 4 for strain ff112 with pMCR-1(PN23)) and **d** ($t = 3.801$, d.f = 4) were calculated by Student's $t$ test

the *G. mellonella* infection model. We chose four non-MCRPEC clinical strains, belonging to different ST types: ST638 (ff112), ST589 (ff141), ST127 (ff156) and ST648 (CX37), which is one of diverse ST groups associated with *mcr-1* gene in China[10]. Three different *mcr-1* plasmids isolated from MCRPEC strains (PN16, PN23 and ZJ487) were transferred to these non-MCRPEC strains by electroporation. We have shown that the acquisition of *mcr-1* naturally occurring plasmids do not affect bacterial growth of the host (wild-type clinical strains) strain and so their fitness cost is modest (Supplementary Fig. 11). However, their virulence is markedly depleted (Fig. 9) supporting our hypothesis that the loss of virulence shown by MCRPEC is not due to growth rate but by the activity of *mcr-1* on the *E. coli* outer membrane (Fig. 3).

**Sequencing analysis of *mcr-1* genetic contexts.** To elucidate genetic changes during the evolution of high levels of colistin resistance, whole-genomic sequencing was performed on all seven wild-type parental strains (day 0) and their isogenic HLCRMs (day 14). For each strain, a read coverage of at least 80-fold was generated and trimmed using Trim Galore and the genomes were de novo-assembled into contigs using SPAdes (3.9.0) with predefined kmers set (Methods). When comparing genomes with their parental strains, the seven HLCRMs showed no amino acid mutations in *mcr-1*-carring contigs including their promoter and the immediate surrounding genetic contexts. In addition, they all possess *mcr-1*-carrying plasmids, exception of strain PN43, which *mcr-1* gene located in chromosome (Supplementary Table 1 and Supplementary Fig. 5). As shown in Supplementary Fig. 1A, there are three representative genetic contexts of *mcr-1*-carrying contigs were identified in seven MCRPEC. Firstly, the *mcr-1* identified on chromosome in PN43 isolate was flanked directly by IS*apl1*, the genetic context of which is identical to other chromosomal *mcr-1* gene (accession number: KY421935, LT594504 and ENQ00000000), except that a flanking hypothetical open-reading frame is truncated by IS*1294*. Of note, in two IncI2 plasmids from PN16 and PN21 isolates, the genetic context of *mcr-1* are identical to pHNSHP45 (accession number: KP347127) harbouring

IncI2 scaffold and type VI secretion system components (Supplementary Fig. 1B). The rest of four *mcr-1* positive IncX4 plasmids from PN23, PN24, PN25 and PN42 isolates exhibit remarkable similarity to the backbone of pMCR-NJ-IncX4 (accession number: KX447768) (Supplementary Fig. 1C).

## Discussion

Acquiring antibiotic resistance by mutation or horizontal gene transfer tends to be associated with a fitness cost[26–28]. This cost plays a key in limiting the spread and maintenance of resistance in pathogen populations by generating selection against resistant strains under conditions where antibiotic doses are low, for example, during transmission between hosts[28,29]. However, recent studies have challenged this paradigm and provide evidence supporting the notion that, at least in some systems, the evolution of resistance can be associated with fitness advantages including the enhanced ability to cause disease[30,31,32]. We have shown that the case for *mcr-1* imposing a fitness cost on its *E. coli* host is substantial and increased expression of *mcr-1* not only impairs cell growth (Fig. 2a) and diminishes bacterial fitness (Fig. 2c), but also alters the cell's architecture (Fig. 3c) and kills the bacteria (Fig. 4c). Evolving colistin resistance by acquiring *mcr-1* therefore challenges bacterial populations with an evolutionary trade-off: high-level expression of *mcr-1* provide protection against the antibiotic, but this increase in resistance compromises growth rate, fitness, membrane structural integrity and increases cellular death. This trade-off may explain several important phenomena. First, *mcr-1*-mediated colistin MICs are at best moderate (usually 2–8 mg l$^{-1}$) when compared to colistin resistance (8–256 mg l$^{-1}$) mediated by *pmrA/pmrB*[3,9,10,16]. Secondly, we show that *mcr-1* plasmid copy number is very low when compared to other plasmid-mediated mechanisms, eg, IncA/C plasmids carrying *bla*$_{NDM-1}$ and this appears to protect the cell from too many copies of *mcr-1*[21,33].

As a negative control, we show that the high expression of *bla*$_{TEM-1}$ has little, if any, effect on *E. coli* fitness and survival

(Fig. 5c, Supplementary Figs. 3C and 4C), and therefore the effects witnessed with MCR-1 are due to its incorporation into the *E. coli* membrane or by its catalytic action—the phosphoethanolamine modification of LPS. It is interesting to note that only a few mobile membrane-associated antibiotic resistance mechanisms have been characterised. One of them is the tetracycline efflux system, eg, *tetA(B)*, which is tightly regulated by its neighbouring gene, *tetR*. It has been shown that overproduction of *tetA* is lethal, which is probably due to its effect on the membrane topology[34–36]. Unlike the *tetA*-like efflux system, *mcr-1* does not possess an adjacent repressor to control its expression and its expression is controlled by other regulatory systems such as low plasmid copy number. Compared with the considerable burden of high expression of MCR-1 (Figs. 2–4), the toxic effects of its two derivatives, MCR-1 E246A and MCR-1 soluble region, are moderate, which can be observed by some restoration of its outer membrane integrity and a reduction in death rate (Fig. 5, Supplementary Figs. 3 and 4). However, as the toxic effects are more evident in MCR-1 E246A than the MCR-1 soluble region, MCR-1's cellular burden is due to both the embedding of the protein in the *E. coli* outer membrane and MCR-1's phosphoethanolamine modification of LPS.

China has now withdrawn colistin as an animal growth promoter and is now being deployed in human medicine[4,37]. However, the notion that MCRPEC will drastically diminish on withdrawal of colistin is parsimonious and naive. Our data previously, and the stability data on HLCRMs herein, suggests that colistin resistance in MCRPEC and HLCRMs can be stable[9]. Additionally, our data on *E. coli* TOP10 (*mcr-1*/pBAD) shows that HLCRMs can be generated from a lab-strain, but only after acquisition of *mcr-1* suggesting the increase in colistin resistance is not dependent on the strain background that is further supported by the fact that all our selected wild-type MRCPECs readily generated HLCRMs. Finally, it is possible that colistin-resistant strains will evolve compensatory adaptations that allow for *mcr-1* to be expressed at high levels at a low fitness cost. Compensatory adaptation is routinely detected in vitro[28], and there is some evidence that compensatory adaptation maintains otherwise costly resistance mutations in vivo[38].

Due to its central clinical role in causing Gram-negative bacterial sepsis, lipid A as an immune modulator has been thoroughly scrutinised[39,40]. However, despite the fact that the mechanisms of 4-amino-4-deoxy-L-arabinose and phosphoethanolamine LPS modification are well known, there have been very few systematic studies examining their impact on lipid A as an immune stimulant. Studies on *Salmonella* have examined the addition of phosphoethanolamine and found no significant change in virulence[41]. These data are supported by a recent study in *Haemophilus ducreyi* that used knockout deletions of phosphoethanolamine modifying genes (*lptA*, *ptdA* and *ptdB*) and concluded that the triple mutant was as virulent as the parent[42]. A recent study by John et al.[43] examined the phosphoethanolamine and its effect in *Neisseria meningitidis* by comparing invasive and carrier isolates and found more phosphoethanolamine and sialic acid substitutions from invasive strains suggesting that phosphoethanolamine-lipid A modification enhances virulence. This finding also has been identified in non-*mcr-1* mediated colistin resistance in *K. pneumoniae*, where lipid A remodelling mediated by the *mgrB* mutation, resulted in increased colistin resistance enhancing the virulence of *K. pneumoniae* by decreasing the affinity of colistin and attenuating host defence response[44]. These data are further supported by O'Brien and colleagues examining *Campylobacter jejuni* phosphoethanolamine modified lipid A, showed increased recognition of a human Toll-like receptor and increased commensal colonisation in mice[45]. However, Zughaier et al.[46] examined *N. gonorrhoeae*

phosphoethanolamine modified lipid A and showed that it reduced autophagy in human macrophages and useful mechanism to evade the host immune system. Thus far, there are no similar studies on *E. coli* and this is the first examining the effect of *mcr-1* on *E. coli* fitness and virulence. The data presented in this study indicate that *E. coli* (in this case, *E. coli* TOP10 (*mcr-1*/pBAD)) is forced to finely tune the expression of *mcr-1* or else the overexpression becomes toxic resulting in profound changes in the architecture of the outer membrane (Fig. 3c), causing leakage of cellular cytoplasm (Fig. 3c) and death (Fig. 4c). Interestingly, in a HLCRM (PN21) where that was at least 11-fold expression on *mcr-1* (Supplementary Fig. 8), we observed no cellular deterioration (Fig. 9b, c), suggesting that PN21 may have a different genetic makeup and is able to produce compensatory mutations[47]. When we have examined the immediate *mcr-1* genetic context comparing HLCRMs and their respective wild-type parents, no changes in the promoter regions in any of our seven isogenic pairs could be identified (Supplementary Fig. 1). When compared to their respective wild-type parental strains, most of HLCRMs, except strains PN25 and PN42, showed no amino acid mutations in PmrE, PmrAB, PmrC, PhoPQ, MgrB, ArnB/D, CptA, EptB, IpxM, MicA and ArcAB, which are known to be associated with colistin resistance[3,48]. However, in strains PN25 and PN42, amino acid mutations have been identified in PmrA (R81S)[48] and PmrB (V161M)[49], respectively, which are responsible for reducing susceptibility of colistin. Another finding is that HLCRMs altered the susceptibility to other antimicrobials (Supplementary Table 7). HLCRMs have obtained higher MICs of colistin (16–256 mg l$^{-1}$) by multiple passages in the increasing concentrations of colistin. Conversely, these mutants are more susceptible to other antibiotics (Supplementary Table 7), for instance, expect strain PN24, the other six HLCMRs strains caused one to 16-fold MIC reduction of chloramphenicol, compared with their respective parental strains. The MICs of tigecycline, regarded as one of the last resorts to treat infections caused by multidrug resistance bacteria, have one to 32-fold reduced in all HLCRMs. This is likely due to *mcr-1*-mediated membrane permeability in these strains.

As *mcr-1* continues to spread globally, and the clinical impact is assessed, it will be interesting to examine how virulent MCRPEC compare with non-MCRPEC. In this study, *mcr-1* positive plasmids were transferred to non-MCRPEC clinical strains (ST638 (ff112), ST589 (ff141), ST127 (ff156) and ST648 (CX37)) by electroporation. We have shown that the growth rate of all *mcr-1*-positive transformants appears to be lower than that of their non-MCR parents, but the difference is not statistically significant (Supplementary Fig. 11). Compared to non-MCRPEC strains from phylogenetic group D (considered as second-most virulent ExPEC group), the mortality of *G. mellonella* showed a marked reduction after acquisition of a *mcr-1*-positive plasmid (Fig. 8). To conclude, it would appear the acquisition of *mcr-1* by *E. coli* is a 'poisoned chalice'—on the one hand *mcr-1* is required to provide protection in a colistin-rich environment, yet acquisition compromises the bacterium's normal physiology; furthermore, overexpression results in acute 'toxicity'.

## Methods

**Bacterial strains and growth conditions**. Seven *E. coli* isolates carrying the *mcr-1* gene (MCRPEC) were isolated and characterised from Thailand and used as parental strains (Supplementary Table 1). Bacteria were grown in LB broth or chromogenic agar (Liofilchem, Roseto, Italy), supplemented with appropriate antibiotics at 37 °C. *E. coli* TOP10 (*mcr-1*/pBAD) was constructed by cloning the *mcr-1* gene in-frame into low-copy number pBAD-hisA expression vector (ThermoFisher, UK) such that the expression of *mcr-1* is controlled by the pBAD promoter of the *araBAD* (arabinaose) operon that can be regulated with L-arabinose induction[50]. Four non-MCRPEC clinical strains belonging to *E. coli* group D,

were chosen as recipients for transformation by electroporation. All strains and plasmids used in this study are listed in Supplementary Table 2.

**Plasmid constructs.** An MCR-1 non-active enzyme was created by a single substitution (E246A). Its encoding gene was excised from *mcr-1*-pUC19 (E246A)[19] and sub-cloned into plasmid pBAD-HisA to generate E246A/pBAD. To test the effect of the MCR-1 soluble domain on bacterial growth, the gene encoding the MCR1 soluble domain (lacking the five predicted transmembrane helices; codons 219–541) was cloned into pBAD-HisA vector with forward and reverse primers (*mcr-1*F soluble and *mcr-1*R soluble, Supplementary Table 3). As a negative control, a ß-lactamase gene, *bla*TEM-1b, a 861 bp fragment were generated by PCR with forward and reverse primers (Supplementary Table 3) and then inserted into pBAD-HisA vector using the restriction sites *Eco*RI and *Kpn*I. All resultant plasmids were transformed into *E. coli* TOP10 cell (Invitrogen, UK), purified and its integrity confirmed by PCR, double restriction digestion and DNA sequencing.

**Electroporation.** Overnight cultures were diluted (1:50) in 5 ml fresh LB broth and incubated at 37 °C (220 r.p.m.) until the cultures reached an OD$_{600}$ value of 0.5–0.7. Bacterial cell cultures were kept on ice for 20 min, followed by centrifugation at 5000 r.p.m. (14,000×g), 4 °C for 15 min. The pellets were then resuspended in 1 ml chilled 10% glycerol and re-centrifuged for 15 min at 4 °C. After another two washes in chilled 10% glycerol, supernatants were discarded and the pellets were suspended in the residual glycerol. Subsequently, 50–100 ng of plasmid DNA were added into 50 μl of electrocompetent cells and then the DNA cell mixture was transferred into a chilled cuvette. EC3 (3.0 kv, 5.5 ms) programme was used for electroporation (Bio-Rad MicroPulser, France). Immediately, 950 μl of warm LB broth was added to the cuvette and thoroughly mixed, followed by incubation at 37 °C for 1 h, shaking vigorously at 220 r.p.m. A total of 100 μl aliquot were plated onto a pre-warmed selective plate and incubated overnight at 37 °C.

**Quantitative real-time PCR.** Expression of *mcr-1* was assayed by a two-step qRT-PCR using primers *mcr-1*-qF, *mcr-1*-qR and *mcr-1* probe, with Precision 2× qPCR Mastermix (PrimerDesign, UK) following manufacturers' protocol. Total RNA was extracted from bacteria using the RNeasy Plus kit with on column DNase digestion (Qiagen, Germany), followed by complementary DNA synthesis with DNA-integrated genomic DNA (gDNA) removal using QuantiTect Reverse Transcription kit (Qiagen, Germany), according to manufacturers' protocol. The absence of carry-over gDNA was verified for every experiment by comparative qPCR (or standard PCR) in the absence of reverse transcriptase. *rpoB* expression level was used as internal control using primers *rpoB*-qF, *rpoB*-qR and *rpoB* probe. Relative expression results were obtained by the ΔΔCT analysis method using mean CT value. For details of used primers and probes, see Supplementary Table 3.

For qPCR determination of *mcr-1* copy numbers per cell, 0.1 ng of total gDNA was used as template with primers *mcr-1* qF, *mcr-1* qR and *mcr-1* probe and 16S primers and probe (Supplementary Table 3). In parallel standard curves for *mcr-1*, 16S were obtained using as template serial dilutions of *mcr-1*-carring plasmid DNA extracted from pSU18-*mcr-1* strain[19] (4.3 pg of DNA corresponding to 10$^6$ copies, calculated through the website: http://cels.uri.edu/gsc/cndna.html) and *E. coli* TOP10 (Invitrogen) total gDNA (5 ng corresponding to 10$^6$ cells)[21], respectively.

**Selection of induced high-level colistin-resistant mutants.** HLCRMs were generated from seven wild-type *mcr-1* positive strains through 14-day serial passaging with increasing concentrations of colistin (Alfa Aesar, US). Overnight cultures of parent strains were diluted to 10$^5$ c.f.u. per ml and challenged with different concentrations of colistin (from 0.125 to 256 mg l$^{-1}$) for 18–20 h at 37 °C. The next day, cultures in the last well that yielded visible bacterial growth were mixed with the first clear well (normally registered as the MIC) and challenged with colistin as described above. Cultures at 3, 7, 11 and 14 days were retained and stored at −80 °C for further analysis.

**Stability of high-level colistin resistance in HLCRMs.** To detect whether the colistin resistance in HLCRMs is stable, ie, reversible or not, serial passages of HLCRMs were performed in colistin-free medium. Overnight cultures of HLCRMs were diluted (1:500) into fresh LB broth without colistin and incubated with vigorous shaking (220 r.p.m.) for 18 h. To measure the proportion of colistin resistance bacterial population during reversion, overnight cultures were serial diluted, then inoculated on two types of chromogenic agar: one type is free of antibiotics, the other is containing various concentrations (8, 16, 32, 64, 128, 256 mg l$^{-1}$) of colistin depending on the level of colistin resistance mediated by the HLCRM. The c.f.u. per ml of colistin resistance cells were counted after 18–22 h incubation at 37 °C.

**Confocal laser scanning microscopy imaging using LIVE/DEAD staining.** *E. coli* TOP10 (*mcr-1*/pBAD), *E. coli* TOP10 (pBAD only) and *E. coli* TOP10 (*n* = 4) were grown overnight in LB broth supplemented with 100 mg l$^{-1}$ ampicillin (Fisher Chemical, UK) at 37 °C (120 r.p.m.). Overnight cultures were standardised to OD$_{600}$0.05 and inoculated (1:10; v/v) into 96-well glass-bottomed plates (Whatman®, UK) in LB broth for 16 h (37 °C; 30 r.p.m.). The supernatant was gently removed and the biofilms were further incubated in fresh LB broth ± L-arabinose (0.2%, w/v) for 8 h. The supernatant was removed and the biofilms stained with 6% LIVE/DEAD® (v/v; BacLight™ Bacterial Viability Kit, Invitrogen)

in phosphate buffered saline (PBS) prior to CLSM imaging (Leica TCS SP5) with ax63 lens. The CLSM *z*-stack images were analysed using COMSTAT image analysis software for quantification of biofilm biomass[51]. The COMSTAT data was assessed using one-way ANOVA followed by Tukey–Kramer Multiple comparisons post hoc test. Statistical significance was set at *p* < 0.05.

**In vitro competition assays.** In vitro competition experiments were used to measure the relative fitness of the *mcr-1* HLCRMs, *E. coli* TOP10 (*mcr-1*/pBAD) and *E. coli* TOP10 (pBAD only) and *E. coli* TOP10. These strains were competed against a GFP-labelled *E. coli* DH5-alpha carrying plasmid pHT315-pAphA3′-gfp for constitutive expression[52] and flow cytometry was used to measure changes in the cell titre of the two strains during competition. All competitions were carried out in M9 medium (Sigma-Aldrich, UK) with six replicates per strain/condition, as previously described with some modifications[53]. For *E. coli* TOP10 (*mcr-1*/pBAD), *mcr-1* expression was induced in a controlled way by adding different concentrations of L-arabinose (0, 0.0002, 0.002, 0.02 and 0.2%, w/v).

The bacteria were cultured overnight in LB supplemented with the appropriate antibiotics (2 mg l$^{-1}$ of colistin for HLCRMs, or 100 mg l$^{-1}$ of ampicillin (Fisher Chemical) for *E. coli* TOP10 (*mcr-1*/pBAD), *E. coli* TOP10 (pBAD) and the GFP-labelled *E. coli* DH5-alpha. The overnight cultures were diluted 1:400 in M9 broth and mixed at 1:1 ratio with GFP-labelled cells. Before starting the competitions, the exact initial proportion of fluorescent/non-fluorescent cells was estimated using flow cytometry (for details see below). If the actual ratio was close to 1:1, we started the competition by shifting the mixtures to a shaking incubator (37 °C, 225 r.p.m.). Otherwise, the preparation procedure was repeated.

After 22 h (6 h for HLCRMs), the competed bacteria were diluted 1:400 in M9 and analysed on a flow cytometer to estimate the resulting proportion of labelled vs unlabelled cells. Flow cytometry was performed on an Accuri C6 (Becton Dickenson, Biosciences, UK). The cell densities were adjusted to give ~1000 events per microlitre. During data acquisition, a lower cut off was set at 10,000 for FSC-H and at 8000 for SSC-H. For each competition, we ensured that the GFP-labelled strain can be well separated from non-fluorescent strains by comparing non-mixed controls (overlap is usually <2% of the cells). Relative fitness was calculated using formula:

$$\text{Relative\_fitness} = \frac{\log_2\left(\frac{p_1}{p_0/n_{\text{dilution}}}\right)}{\log_2\left(\frac{1-p_1}{(1-p_0)/n_{\text{dilution}}}\right)},$$

where $p_0$ is an initial proportion of an unlabelled stain, and $p_1$ is a final proportion of an unlabelled stain after competition. The $n_{\text{dilution}}$ is the factor, which reflects a fold difference in cell density at the beginning and at the end of the competition. For HLCRMs, we expressed the fitness of daughter strains relative to their parental strains (ie, $f_{\text{daughter}}/f_{\text{parental}}$) and followed the procedure of error propagation to account for the uncertainty of the two estimates:

$$\text{SE} = \sqrt{\left(\frac{\text{SD}_{\text{daughter}}}{\overline{f}_{\text{daughter}}}\right)^2 + \left(\frac{\text{SD}_{\text{parental}}}{\overline{f}_{\text{parental}}}\right)^2},$$

where $\overline{f}$ and SD are a mean estimate and its SD for each corresponding strain based on six replicates. Similarly, the relative fitness of the HLCRMs at different L-arabinose concentrations was represented as the relative fitness at no induction (no L-arabinose). The differences in fitness were tested using non-parametric Mann–Whitney test, and the *p* values were adjusted by Bonferroni method.

**LPS isolation and macrophage stimulation.** LPS was extracted using LPS extraction kit (iNtRON Biotechnology, UK), according to the manufacturers' instructions. To eliminate protein contamination, treatment with protease K was performed prior to the extraction steps. The protease K (AppliChem, Germany) (20 mg/ml) was added to the cell mixture and incubated at 56 °C for 1 h. The efficiency of the LPS preparation was determined by Limulus Amebocyte Lysate (LAL chromogenic Endotoxin Quantitation kit, ThermoFisher) to measure the LPS concentration, according to the manufacturers' instructions.

THP-1 human cells (American Type Culture Collection, ATCC TIB-202™) were cultured with the medium (RPMI-1640 medium supplemented with 10 % foetal calf serum, HEPES, L-glutamine, ampicillin and streptomycin) (Sigma-Aldrich) in a humidified cell culture incubator with 5% CO$_2$ at 37 °C. THP-1 cells were differentiated into macrophage cells by stimulation with phorbol myristate acetate (PMA). Cells were centrifuged and resuspended in fresh supplemented RPMI-1604 medium to a concentration of $8 \times 10^5$ cells per ml. Phorbol myristate acetate (10 ng ml$^{-1}$) was added to diluted cells, and then aliquoted into 24-well plate by adding 1 ml of cell suspension per well. The differentiation of THP-1 monocytes were complete after 48 h of incubation, as exemplified by induction of adherent cells phenotype under the microscope.

Phorbol myristate acetate and any non-adherent cells were removed by replacement of the medium with fresh medium prior to addition of serial concentrations of LPS (4.5, 0.45, 0.045, 0.0045, 0.00045 ng/ml) to the appropriate wells in triplicate. Differentiated THP-1 cell cultures without LPS served as the negative controls. The wells are thoroughly mixed and incubated at 37 °C for 24 h. The samples were collected at 4, 6, 8 and 24 h and stored at −20 °C. The production of macrophage-derived cytokines (IL-6 and TNF-alpha) were analysed using DuoSet® ELISA kit (R&D systems), and cytokine concentrations were calculated according to the manufacturers' instructions.

**Morphological analysis by TEM**. Overnight cultures were diluted into 50 ml of fresh media supplemented with 100 mg l$^{-1}$ ampicillin for *mcr-1*/pBAD and 2 mg l$^{-1}$ colistin for PN21 strains, respectively. For *mcr-1*/pBAD strain, 0.2% of L-arabinose was added to induce the overexpression of *mcr-1*. After 8 h incubation, samples were fixed by addition of glutaraldehyde to the broth to a final concentration of 1%. Bacteria were harvested by collection onto 0.45 mm pore filters, gently scraped off and dispersed in 4% low melting point agarose at 50 °C. Preparations were allowed to gel at room temperature and cut into 1 mm cubes. Cubes were post-fixed for 2 h in 2% uranyl acetate, washed for 3 × 20 min in reverse osmosis purified water and dehydrated through graded propan-2-ol (50%, 70%, 90% for 10 min each, 100% for 2 × 15 min), infiltrated with LR white acrylic resin (London Resin Company, Aldermaston, UK) (50% in propan-2-ol 30 for min, neat resin for 4 × 20 min) placed in size 0 gelatine capsules with fresh resin and heat polymerised overnight at 50 °C. Thin (80 nm) sections were cut on an Ultracut E. ultramicrotome with a glass knife and collected onto 300 mesh copper grids, stained with lead citrate and examined in a Philips CM12 (FEI UK Ltd. UK) TEM at 80 kV. Digital images were captured with a Megaview III digital camera and AnalySIS (Soft Imaging System GmbH, Germany).

***Galleria mellonella* infection model**. In vivo virulence of seven wild-type MCRPEC isolates and their HLCRMs (day 0, 3, 7, 11 and 14) were evaluated using *G. mellonella* infection model. The wax moth *G.mellonella* in larval stage (Live Foods UK Ltd., http://www.livefood.co.uk) were stored in dark and used within 3 days from shipment. Prior to inoculation into larvae, bacterial pellets were washed with sterile saline and then diluted to an appropriate cell density. Using a 50 µl Hamilton syringe, 10 µl aliquots of serially diluted bacterial suspension (from 10$^3$ to 10$^7$ c.f.u. per ml) were injected into the haemocoel of each larvae, through the rear left pro-leg[54–57]. A group of 10 larvae were randomly chosen to inject for each level of inoculation in triplicate. Followed by injection, larvae were incubated at 37 °C, and the survival of larvae was monitored daily for 3 days. Death was denoted when larvae no longer responded to touch. Results were analysed by Kaplan–Meier survival curves (GraphPad Prism statistics software). For all experiments, three control groups were used: 10 larvae in first groups were injected with 10 µl sterile saline, the second group included larvae that received mock injection to ensure death was not caused by physical trauma, and the larvae in third group with no injection. In all cases, no dead larvae were observed in the control groups.

**S1 nuclease-based pulsed-field gel electrophoresis**. To investigate whether *mcr-1* genes were located on plasmids or chromosome, endonuclease S1 pulsed-field gel electrophoresis were assayed as previously described[58]. Briefly, bacterial DNA was prepared in agarose blocks and digested with 1 unit of S1 nuclease (Invitrogen). Electrophoresis was conducted on a CHEF-DR III apparatus (Bio-Rad, Hercules, CA, USA) under the following conditions: 6 V cm$^{-1}$ at 14 °C, with an initial pulse time of 4 s and a final pulse time of 45 s for 18 h. In-gel hybridisation was done with a *mcr-1* probe labelled with $^{32}$P (Stratgene, Amsterdam, Netherlands) with a random primer method according to manufacturers' instruction.

**Whole-genome sequencing and bioinformatics analysis**. Total gDNA was extracted from an overnight culture (2 ml) on a QIAcube automated system (Qiagen). Following extraction, gDNA was quantified by fluorometric methods using a Qubit (ThermoFisher Scientific), with quality ratios of gDNA (A260/280 and 260/230) determined via Nanodrop (ThermoFisher Scientific). Genomic DNA libraries are prepared for whole-genome sequencing using the NexteraXT kit (Illumina), as described by the manufacturer. Paired end sequencing was performed using the Illumina MiSeq platform (MiSeq Reagent V3 Kit; 2 × 300 cycles). For each *E. coli* isolate, at least 80× coverage was generated. Raw sequence reads were trimmed using Trim Galore and the genomes were de novo-assembled into contigs using SPAdes (3.9.0) with pre-defined kmers set. Raw reads were also assembled with Geneious (10.0.9; Biomatters Ltd.) de novo assembler, set at medium sensitivity for analysis of paired Illumina reads. Geneious was used to map both sets of contigs to reference genes identified by closest BLAST homology and was also used to annotate genes from closest homologues in NCBI Genome database. Resistance genes were identified using Resfinder within CGE[59], and wgMLST profiles were generated using the CGE platform coupled with the PubMLST.org database[60]. Plasmids were identified within the genome assembly and typed using Plasmidfinder[61].

**Analysis of lipid A modifications**. A total of 200 ml overnight culture were harvested by centrifugation at 10,000×*g* for 10 min. Lipid A was extracted as previously described[62]. In brief, the pellets were washed twice and resuspended in 1× PBS, followed by adding single-phase Bligh-dyer solvent (chloroform: methanol: water; 1:2:0.8, v/v/v). The mixtures were incubated for 20 min at room temperature and centrifuged at 2000×*g* for 20 min. The LPS pellets washed with the single-phase Bligh-Dyer solvent and resuspended in hydrolysis buffer by boiling the samples in 50 mM sodium acetate (pH 4.5). The samples were sonicated twice (20 s per burst) and incubated the samples in a boiling water bath for 30 min. Then the lipid A was extracted using a two-phase Blign-Dyer solvent (chloroform:methanol: water; 2:2:1.8, v/v/v). Once extracted, 10 µl of the purified Lipid A dissolved in chloroform-methanol (2:1, v/v) were analysed using a ESI-QTOF mass spectrometery (Waters Synapt HR-MS) in the negative-ion mode. The spectral data were used to analyse the structures of bacterial lipid A from MCRPEC (*E. coli* W3110 with plasmid pUC19-*mcr-1*) and non-MCRPEC (*E. coli* W3110 with empty vector pUC19)[9].

**Data availability**. Genomic sequences of bacterial strains have been deposited in the NCBI GenBank with accession codes MG489944, MG557851, MG557852, MG557853, and MG557854; there are two sequencing data still outstanding. Other relevant data supporting the findings of the study are available in this article and its Supplementary Information files, or from the corresponding authors upon request.

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

## Acknowledgements

We thank Dr Jonathan Tyrell (Division of Infection and Immunity, Cardiff University) for useful suggestions on *Galleria* infection model, Dr M. Toleman (Division of Infection and Immunity, Cardiff University) for his valuable advice on genome sequencing, and Professor W. Jiang (Cardiff China Medical Research Collaborative, Cardiff University) for access to qPCR machine (Applied Biosystems™ StepOnePlus®, UK). This work was supported by MRC grant DETER-XDR-CHINA (MR/P007295/1). Q.Y. is funded by a CSC Scholarship. D.O.A. benefits a Geneva University Hospitals (HUG) and Swiss National Science Foundation (P300PB_171601) overseas fellowship. P.N. and U.T. are funded by Royal Golden Jubilee-PhD Program from Thailand Research Fund and Rajamangala University of Technology Lanna (PHD/0054/2555).

## Author contributions

Conceived and designed the experiment: T.W., J.S., O.B.S. and Ji.S. Performed the experiments: Q.Y., M.L., O.B.S., D.O.A., K.S., E.P., L.P., M.P., P.N. and U.T. Analysed the data: Q.Y., O.B.S., D.O.A., P.H., A.P., Y.W., S.W., Y.S., C.M. and D.T. Writing of the main paper: T.W. and Q.Y. All authors contributed extensively to the work presented in this paper. All authors discussed the results and implications and commented on the manuscript at all stages.

## Additional information

**Competing interests:** The authors declare no competing financial interests.

