## [Peer Review File · Nature Communications]

Reviewers' comments:

Reviewer #1 (Remarks to the Author):

Summary

In the submitted manuscript by Yang et al., the authors explore the virulence and fitness of *Escherichia coli* harboring *mcr-1* in different genomic contexts. In a series of fitness experiments, the overexpression of *mcr-1* and cellular outcomes evaluated (live/dead microscopy and growth). To further understand the impact on pathogenesis, the team also analyzes the effects of overexpression via electron microscopy. The virulence of different clinical isolates before (parent) and after colistin exposure was also determined for different *E. coli* strains that harbor *mcr-1* using a *Galleria mellonella* model of infection. There is some significance tied to this work given the clinical relevance of *mcr-1* and the potential for the spread of colistin-resistance in the global healthcare system. It seems as though it is only a matter of time before certain bacterial species evolve more virulent strains that harbor genes such as *mcr-1* leading to more significant outbreaks. The manuscript is well written for the most part, and the majority of the data is certainly of interest. That said, there are some major and minor concerns that need to be addressed by the authors.

Major concerns

1) The manuscript lacks some key data. There are some extra figures that are not required in the main body of the paper that distract from the overall results. Figures 1, 4, 5, 6, and 8 are the key figures. The others are essentially descriptive or an evaluation of tools that were required to reach answers. Therefore, these extra figures (2, 3, 7, 9) could all be moved to the Supplemental Material. The remaining figures when evaluated in this context appear less convincing in terms of virulence impact and the conclusions being drawn, and this is where the addition of a mammalian model and the use of a clinical isolate as opposed to Top10 for pBAD expression (see below) would strengthen the manuscript and justify the conclusions attempting to be made.

2) Terminology is important. To be clear, the difference between fitness and virulence is critical with regard to this study. Fitness can be used to define any condition that affects bacterial growth, either *in vitro* or *in vivo*. However, virulence specifically refers to how the bacteria attack a host resulting in survival (time to death) results within said host. Pathogenesis is a term used to describe how bacteria proceed along an infection course and can refer ultimately to the demise of a given animal with respect to this infection.

Thus, a strain can be pathogenic, resulting in some markers of infection and disease, but not necessarily virulent (killing the animal). The distinction is very important with regard to wording in the Discussion. The authors cite a number of studies from Lines 375-392. However, an important distinction between these studies and the authors' work is that the cited studies look at the deletion of gene in murine models of infection, and this gene deletion did not affect the overall growth of the organism. Therefore, the work within this manuscript cannot be directly compared to this cited work. The authors need to re-evaluate the Discussion to what conclusions can actually be drawn. If *mcr-1* was deleted from the plasmid, the plasmid was cured, or deleted from the genome (PN42 strain), and hyper-virulence was observed, then the cited-works would be comparable. The authors should consider doing this experiment. The authors should also consider mouse models once mutants are generated as these papers did (see below).

3) *Galleria mellonella* is certainly a well-known, validated means to evaluate virulence. However, before broader statements about virulence and pathogenesis are made with respect to patient outcomes, the results need to also be recapitulated in a mammalian animal model where the pathogen of interest goes through similar steps of pathogenesis leading to an infection seen in humans in order speculate on clinical importance (Lines 402-407). While the *G. mellonella* paper by Peleg et al. is correctly cited. An additional citation where *G.*

mellonella is specifically used to evaluate *E*. *coli* virulence should also be cited (for example EXPEC – Williamson et al. 2015). However, it is extremely important to acknowledge if there are growth/fitness defects *in vitro* (Fig. S3), these strains should not be tested in an animal system at all. There is no doubt they would be attenuated and less virulent than parent strains since they do not replicate as fast even *in vitro*, which is a flaw with regard to the methods used here to evaluate the impact of *mcr-1* on virulence.

4) Figure 3, Figure 4, Figure 8 are of concern because proper controls are not employed. The overexpression of any gene can lead to possible fitness costs. Why are these detrimental effects *mcr-1* specific? With these figures the authors are only comparing pBAD::*mcr-1* to an empty pBAD vector. Overexpression of a housekeeping gene should also be included as one control, and the overexpression of another antibiotic resistance gene should also be considered, especially where the literature could be used to backup these control selections. Also, the pBAD plasmid has a notoriously leaky promoter. Glucose should be used in absence of arabinose at all times to help repress the promoter until the time of the assay (i.e. addition of arabinose). If glucose is used in the media before the addition of arabinose, it is likely the cells will tolerate *mcr-1* better (see Fig. 5A where cells are dying even uninduced). Expression with another system, less leaky, like pTac should be considered. The use of a Tn7 or Tn5 insertion (one copy) could even be considered given the PN42 strain with a single copy on its genome. Meaning, a knock-in that does not affect bacterial growth, but still impacts virulence is more desirable. Similarly, a knock-out (Point#2) is more desirable than the methods used here.

5) Another methods criticism is strain choice. The lab strain, *E. coli* Top10, is used as a surrogate here to show the impact of *mcr-1* and to generate a HLCRM, but *mcr-1* has been isolated from *Klebsiella*, *Enterobacter*, *Citrobacter*, *Salmonella* species, and clinical isolates of *E. coli*. The difference between all of these clinical isolates and the Top10 strain is that they are virulent in animal models, and likely have modified membranes ready to accommodate proteins associated with virulence and antibiotic resistance genes such as *mcr-1*. It is understood the attempt is to show even a lab strain can become an HLCRM, but ultimately, the goal of this work is to understand the impact of *mcr-1* on virulence. For confirmation of virulence results, the pBAD plasmids should be transformed into a clinical isolate, which may or may not handle the presence of *mcr-1*. If this clinical strain can handle the pBAD insertion/expression better, it invalidates the hypothesis of the work presented. If it cannot and the results are similar to the Top10 strain, it further validates the work presented.

6) It has been shown in *A. baumannii* and *K. pneumoniae* that colistin-resistant strains can evolve to retain their virulence (Wand et al. 2015, Durante-Mangoni et al. 2015, Jones et al. 2017, and Kidd et al. 2017). While the colistin-resistance is not *mcr-1* mediated, but *pmrAB* or *mgrB* mediated instead, these papers should at least be cited in the discussion to show it is possible to have compensatory mutation in other species that retain, evolve to parent strain levels of virulence, or even lead to hyper-virulent strains (Kidd et al. 2017). These papers highlight how clinical isolates can evolve to accommodate LPS modification, and it is also in contrast to what is being presented in this submission. Meaning, compensatory mutants will likely evolve in the wild and likely have already evolved despite any negative impact of *mcr-1* expression on bacterial growth, which makes the results and conclusions being drawn by this manuscript less important. Therefore, not sure the data provides “hope” or not with regard to the clinical situation (Lines 399-407). Also, conveying “hope” seems incorrect when people have died when infected with strains containing *mcr-1*. Apparently, the clinical isolates are virulent enough to cause disease and mortality in patients so the conclusion being drawn from this work are out of place.

7) Finally, the authors should evaluate the membrane proteins, lipid content, and phosphoethanolamine addition via mass spec or other types of analysis to show that indeed these modifications are linked to *mcr-1* activity and not artificial with respect to overexpression

or serial passage. Non-specific activity and membrane rearrangement could account for differences seen, and not be directly linked to *mcr-1* expression.

Minor Concerns

1) In the Introduction (Lines 64-67), it is pretty clear that the "spread" of *mcr-1* is linked to the increased surveillance. Clinical labs are doing well as far as tracking something that would be a tremendous threat to patients and caregivers. Therefore, the wording should reflect this and not say it is "uncertain". Instead, it should say something to the effect of "likely due to increased surveillance".

2) Line 76-81 – The impact of *mcr-1* with respect to clinical data is difficult to reconcile, and here the Introduction should reflect that. While mortality rates could fluctuate, the patient status is rarely taken into account. The more compromised a patient is (immunocompromised or one or more comorbidities), the more likely a patient could suffer mortality from bacterial infection. So while mortality rates could be higher, it is truly difficult to tie this to the virulence of any given strain as the patient's condition can vary. This point should at least be mentioned.

3) Line 113-115 – Instead of listing all of these genes, found in the strain, it is better to just say "numerous antibiotic resistance genes" and cite the table in the supplemental material.

4) Line 134 – the units, hours ("h") should have a space between it and number "8".

5) Line 413-414 – LB should be referred to as lysogeny broth...not Luria-Bertani. Bertani himself said so (Ezraty et al. 2014).

6) Is the same scale being used for each TEM picture in Fig. 4? B-2 and C-2 look more zoomed in when compared to A-2.

7) Fig. 9A – *Mellonella* should not be capitalized, should be "*mellonella*"

Reviewer #2 (Remarks to the Author):

In this manuscript Yang et al. provide data to support their claim that the expression of *mcr-1* in *E. coli* results in reduced bacterial fitness, and the authors thus hypothesize that the expression of *mcr-1* must be tightly regulated in order to strike an equilibrium between maintaining colistin resistance and bacterial fitness. Specifically, the authors provide evidence that *mcr-1* expression and over-expression reduces bacterial growth rate, viability and fitness in direct competition experiments. They also provide evidence that *mcr-1* expression results in changes in bacterial cell morphology and that the modifications in lipopolysaccharide that occur due to *mcr-1* expression result in LPS that has less ability to stimulate cytokine release from macrophages. In addition, the authors provide evidence that strains containing the *mcr-1* gene, which typically demonstrate low-level colistin resistance (MICs 2-8), can develop high level colistin resistance (MICs > 64) after selection in the presence of increasing concentrations of colistin. Given the increasing clinical significance of colistin resistance, especially due to *mcr-1* expression, these data are of potential interest to others in the field. In the absence of previous studies performing a detailed analysis of the fitness cost of *mcr-1* expression, these data represent an important degree of novelty. However, there are some methodological details which raise doubts regarding the conclusions drawn by the authors and should be clarified (detailed in Major Comments).

Major Comments.

1. In Figure 3A the authors assess growth effects caused by increasing expression of *mcr-1*. However, the authors begin the experiments with an inoculum of 9.6 log₁₀ CFU of bacteria which

would appear to provide little margin for measuring growth during the exponential phase. Can the authors justify using such a high initial concentration?

2. In Figures 3B and 3C data from control experiments performed in the same way with the strain carrying an empty plasmid (TOP10/pBAD) in order to rule out effects caused by increasing concentrations of arabinose.

3. Do the authors have any evidence that continuing to increase the expression of *mcr-1* beyond a certain point results in additional modification of LPS (for example, does increasing the expression of *mcr-1* from 358-fold a 1132-fold, for example in Figure 3B, increase the addition of ethanolamine to LPS)? If not, the growth and fitness defects observed upon additional expression may simply be due to the additional energy expenditure necessary for the overexpression of a protein.

4. In Figure 4 the authors should provide images from the strain containing the empty plasmid (TOP10/pBAD) in the presence of arabinose in order to rule out any morphological changes due to high arabinose concentrations.

5. In the data presented in Figure 6 it is unclear why different graphs are used for presenting data for IL-6. Can the authors provide a justification. Additionally, why are the same time points and LPS concentrations that were used for measuring IL-6 concentrations not shown for TNF- α concentrations.

6. A major question regarding the data presented in this manuscript is the reason for the increased colistin resistance seen in strains containing *mcr-1*. After selection of these strains in the presence of colistin and observing dramatic increases in MIC values, the authors state that they detect no changes in the expression of *mcr-1* (except in one case), and no mutations in *mcr-1* or surrounding sequences. Could there be mutations in other determinants of colistin resistance (e.g. *pmrA/pmrB*). What is the explanation for the phenotypic changes in these strains?

7. The authors use the *Galleria mellonella* model to assess decreases in pathogenicity of the *mcr-1* expressing strains. Although this model has been validated for assessing fitness of bacterial strains, is it the most appropriate model given the fact that the authors claim that one of the main differences between strains expressing *mcr-1* and those without *mcr-1* is the ability to stimulate the immune response?

Minor Comments.

1. The title is somewhat confusing as it is not clear what "essential cellular defence mechanisms" refers to.

2. There are a number of figures which may not be necessary for supporting the author's claims, for examples, Figure 1B, Figure 2 B and C, and Supplementary Figure 1.

3. In the discussion the authors claim that their data suggest that increased leakage of cellular cytoplasm occurs upon expression of *mcr-1* (line 397). It is unclear how the data presented in Figure 4 support this conclusion.

4. The manuscript contains a number of typographical errors and would benefit from improving the level of English that is used.

Responses to comments raised by reviewers:

We would like to thank you for the pertinent comments and we have edited the manuscript accordingly. We have complied with comments where possible and where we have not been able to fully meet your concerns have provided a coherent and logical counterargument.

In particular, we have provided compelling evidence to confirm the fitness and virulence of *mcr-1* mediated colistin resistance, by comparing MCRPEC and non-MCRPEC clinical strains. Additionally, we have also provided unrequested data that we felt would make a more complete picture on the impact of *mcr-1* and increase the standard of the article. The sentences and paragraphs added to make corrections in the revised manuscript are highlighted in red.

In the following we go through the comments by the reviewers point by point.

Reviewers' comments:

Reviewer #1 (Remarks to the Author):

Summary

In the submitted manuscript by Yang et al., the authors explore the virulence and fitness of *Escherichia coli* harboring *mcr-1* in different genomic contexts. In a series of fitness experiments, the overexpression of *mcr-1* and cellular outcomes evaluated (live/dead microscopy and growth). To further understand the impact on pathogenesis, the team also analyzes the effects of overexpression via electron microscopy. The virulence of different clinical isolates before (parent) and after colistin exposure was also determined for different *E. coli* strains that harbor *mcr-1* using a *Galleria mellonella* model of infection. There is some significance tied to this work given the clinical relevance of *mcr-1* and the potential for the spread of colistin-resistance in the global healthcare system. It seems as though it is only a matter of time before certain bacterial species evolve more virulent strains that harbor genes such as *mcr-1* leading to more significant outbreaks. The manuscript is well written for the most part, and the majority of the data is certainly of interest. That said, there are some major and minor concerns that need to be addressed by the authors.

Major concerns

1) The manuscript lacks some key data. There are some extra figures that are not required in the main body of the paper that distract from the overall results. Figures 1, 4, 5, 6, and 8 are the key figures. The others are essentially descriptive or an evaluation of tools that were required to reach answers. Therefore, these extra figures (2, 3, 7, 9) could all be moved to the Supplemental Material. The remaining figures when evaluated in this context appear less convincing in terms of virulence impact and the conclusions being drawn, and this is where the addition of a mammalian model and the use of a clinical isolate as opposed to Top10 for pBAD expression (see below) would strengthen the manuscript and justify the conclusions attempting to be made.

Reply. We agree with the majority of this reviewer's suggestions and have revised the manuscript accordingly. In order to offer more compelling evidence for the impact of MCR-1 expression on the virulence and fitness, we have transferred *mcr-1* positive plasmids into non-MCRPEC human clinical strains as suggested by this reviewer. Our data shows that the new *mcr-1*-positive transformants (MCRPECT) confers not only slow growth (Fig.S9), but also statistically significant reduced bacterial virulence (changed from "pathogenicity" as suggested by this reviewer) by increasing the survival rate in a *G. mellonella* infection model (Fig.7). These data further confirms that the virulence loss is due to the expression of *mcr-1* gene.

In accordance with this reviewer's suggestions, we have moved figures 2 and 7 to the supplementary materials (Fig.S2 and Fig. S5) and a new figure, figure 7 (in-vivo virulence data), have been added in the manuscript. However, we believe that figures 3 and 9 are vital to the integrity of the manuscript and would argue that they should not be moved to supplementary data. Figure 3 (now the new figure 2) shows the evidence of *mcr-1*-mediated fitness cost by decreasing the growth rate (Fig. 2A) and relative fitness (Fig. 2C) which summarises data pertinent to the key messages of the manuscript. Figure 9 (now the Fig.8) illustrates that the HLCRM PN21 has significantly increased *mcr-1* expression (Fig.S6), but its TEM micrographs show no morphological changes (Fig. 8B-C), indicating PN21 has evolved high-level colistin resistance by evolving compensatory adaptations to cope with this fitness burden.

As agreed with the Editor, we will not include the murine models as we believe they will not add any more information than what is already available. Our Galleria data is comprehensive (considerably more than murine models) and very tight with minor error rates, and these have been repeated at least in triplicate with at least ten larvae per dataset. Therefore, the Galleria model facilitates large numbers to ascertain cause and affect without being ethically challenging (as appose to the murine model).

2) Terminology is important. To be clear, the difference between fitness and virulence is critical with regard to this study. Fitness can be used to define any condition that effects bacterial growth, either in vitro or in vivo. However, virulence specifically refers to how the bacteria attack a host resulting in survival (time to death) results within said host. Pathogenesis is a term used to describe how bacteria proceed along an infection course and can refer ultimately to the demise of a given animal with respect to this infection.

Thus, a strain can be pathogenic, resulting in some markers of infection and disease, but not necessarily virulent (killing the animal). The distinction is very important with regard to wording in the Discussion. The authors cite a number of studies from Lines 375-392. However, an important distinction between these studies and the authors' work is that the cited studies look at the deletion of gene in murine models of infection, and this gene deletion did not affect the overall growth of the organism. Therefore, the work within this manuscript cannot be directly compared to this cited work. The authors need to re-evaluate the Discussion to what conclusions can actually be drawn. If *mcr-1* was deleted from the plasmid, the plasmid was cured, or deleted from the genome (PN42 strain), and hyper-virulence was observed, then the cited-works would be comparable. The authors should consider doing this experiment. The authors should also consider mouse models once mutants are generated as these papers did (see below).

Reply: Whilst we do not fully agree with the definitions suggested by the reviewer, we have nonetheless changed the wording throughout the manuscript in accordance with these wishes. We have revised and provided more evidence to justify our conclusions (new Fig. 7 and Fig.S9) but have also tempered our conclusion in accordance with the reviewer's comments. Herein we compared the virulence effect between MCRPEC(T) transformants and their counterpart non-MCRPEC strains from clinical settings. Consistent with our previous data, the once bacterium has acquired a *mcr-1* positive plasmid, MCRPEC(T) confer a striking virulence loss in *G. mellonella* infection model confirming that this phenomena is not restricted to laboratory strains such as *E. coli* TOP10.

Whilst we could delete the *mcr-1* from the plasmid, the fact that we can induce the expression of *mcr-1* alone and show it has a pronounced effect on growth, fitness and virulence is clear evidence of the deleterious impact of *mcr-1* expression.

We fully understand that we cannot directly compare the virulence displayed by the Galleria models to patient-outcome as there are considerable variants and co-morbidities with patients that cannot be accounted for in animal models. However, we feel it would be negligent not to discuss the data from the few clinical studies examining the clinical impact of MCRPEC. In accordance with the reviewers suggestion we have re-written sections of the Discussion to reflect his/her concerns and have toned down that part where we have extrapolated the findings from the Galleria model (lines 432-445).

3) Galleria mellonella is certainly a well-known, validated means to evaluate virulence. However, before broader statements about virulence and pathogenesis are made with respect to patient outcomes, the results need to also be recapitulated in a mammalian animal model where the pathogen of interest goes through similar steps of pathogenesis leading to an infection seen in humans in order to speculate on clinical importance (Lines 402-407). While the *G. mellonella* paper by Peleg et al. is correctly cited. An additional citation where *G. mellonella* is specifically used to evaluate *E. coli* virulence should also be cited (for example EXPEC – Williamson et al. 2015). However, it is extremely important to acknowledge if there are growth/fitness defects in vitro (Fig. S3), these strains should not be tested in an animal system at all. There is no doubt they would be attenuated and less virulent than parent strains since they do not replicate as fast even in vitro, which is a flaw with regard to the methods used here to evaluate the impact of *mcr-1* on virulence.

Reply: The recommended reference has been added (lines 622). Virulence comparisons between MCRPEC (T) and their counterparts non-MCRPEC clinical strains belonging second-most virulent *E.coli* group (group D), have now been conducted as suggested by this reviewer. By contrast, MCRPEC(T) confer a striking virulence loss in a *G. mellonella* infection-model compared with the wild-type clinical non-MCRPEC strains (Fig.7, lines297-307). We have shown that the acquisition of *mcr-1* naturally occurring plasmids do not affect bacterial growth of the host strain and so their fitness cost is modest (Fig S9). However, their virulence is markedly depleted (Fig. 7) supporting our hypothesis that the loss of virulence shown by MCRPEC is not due to growth rate but by the activity of *mcr-1* on the *E. coli* outer membrane. These findings are also supported by the TEM data on increased MCR-1 expression (Fig. 3C). Therefore, we feel confident to conclude that colistin resistance mediated by *mcr-1* attenuates bacterial virulence and would note that we have purified LPS from MCRPEC and non-MCRPEC and reproducibly shown that the MCR-1 modified LPS does not stimulate immune factors (IL-6 and TNF) compared to the WT LPS in keeping with the rest of the data presented in the manuscript (Fig.5).

4) Figure 3, Figure 4, Figure 8 are of concern because proper controls are not employed. The overexpression of any gene can lead to possible fitness costs. Why are these detrimental effects *mcr-1* specific? With these figures the authors are only comparing pBAD::*mcr-1* to an empty pBAD vector. Overexpression of a housekeeping gene should also be included as one control, and the overexpression of another antibiotic resistance gene should also be considered, especially where the literature could be used to backup these control selections. Also, the pBAD plasmid has a notoriously leaky promoter. Glucose should be used in absence of arabinose at all times to help repress the promoter until the time of the assay (i.e. addition of arabinose). If glucose is used in the media before the addition of arabinose, it is likely the cells will tolerate *mcr-1* better (see Fig. 5A where cells are dying even uninduced). Expression with another system, less leaky, like pTac should be considered. The use of a Tn7 or Tn5 insertion (one copy) could even be considered given the PN42 strain with a single copy on its genome. Meaning, a knock-in that does not affect bacterial growth, but still impacts virulence is more desirable. Similarly, a knock-out (Point#2) is more desirable than the methods used here.

Reply: To address the concerns of this reviewer and in order to rule out any morphological changes and cell viability due to high arabinose concentrations or other genes expression, the control strain containing the empty plasmid (pBAD) in the presence of 0.2% arabinose was analysed by TEM (Fig.S3) and Dead/Live assays (Fig.4B). There are no morphological changes observed in Fig.S3 and therefore the addition of 0.2% arabinose alone possesses little or no toxicity (or even cell death) as seen in Fig.4B. Therefore, it can be reasonably concluded, that the cell wall changes and cell viability reduction are due to the expression of *mcr-1* gene only.

Based on our results (Fig.2A, Fig.2C and Fig.3A), the control *E. coli* TOP10 cell carrying pBAD::*mcr-1* in the absence of arabinose possesses characteristics (growth rate, relative fitness and cell membrane) that are very similar to the negative control, *E. coli* TOP10 cells with pBAD empty vector. We therefore believe we have provided sufficient evidence that the pBAD vector is not “leaky” under these experimental conditions and is appropriate for this study. Additionally, we have now studied the effects of *E. coli* pBAD induced with arabinose without *mcr-1* and shown no effect on cell morphological changes (Fig.S3) and cell viability (Fig.4A).

It is not correct to claim that over-expression of any gene will have a deleterious effect on fitness as this system (pBAD) is the classical archetype model for gene and expression and protein production and having spent 20 years over expressing. For example, β -lactamase genes, I have never witnessed compromised growth even when the gene is expressed 400- to 500 fold (either with IPTG or L-arabinose induction). Furthermore, when we have examined the over-expression of β -lactamase genes in-situ by creating derepressed mutants (in *Aeromonas* and *Stenotrophomonas*), the mutants grow at least as well as the wild-type strains if not better. We would therefore conclude that the bacterium’s response to over-expressing *mcr-1* is atypical, if not unique,

Furthermore, we have added new data the revised manuscript (lines 297-307) to address this reviewer’s concerns. *mcr-1* positive plasmid was transferred into non-MCRPEC clinical strains (one a common Chinese ST group (ST648)) and their virulence assessed. The MCRPEC transformants possess a striking virulence loss when tested in a *G. mellonella* infection model, but their growth rates have very minor changes, indicating that the fitness cost of MCRPEC is modest (Fig S9; Fig. 7). These findings may explain the rapid dissemination of *mcr-1* among wild-type *E. coli* strains in China.

5) Another methods criticism is strain choice. The lab strain, *E. coli* Top10, is used as a surrogate here to show the impact of *mcr-1* and to generate a HLCRM, but *mcr-1* has been isolated from *Klebsiella*, *Enterobacter*, *Citrobacter*, *Salmonella* species, and clinical isolates of *E. coli*. The difference between all of these clinical isolates and the Top10 strain is that they are virulent in animal models, and likely have modified membranes ready to accommodate proteins associated with virulence and antibiotic resistance genes such as *mcr-1*. It is understood the attempt is to show even a lab strain can become an HLCRM, but ultimately, the goal of this work is to understand the impact of *mcr-1* on virulence. For confirmation of virulence results, the pBAD plasmids should be transformed into a clinical isolate, which may or may not handle the presence of *mcr-1*. If this clinical strain can handle the pBAD insertion/expression better, it invalidates the hypothesis of the work presented. If it cannot and the results are similar to the Top10 strain, it further validates the work presented.

Reply: We fully accept this criticism; however, we have generated HLCRMs from 7 wild-type MCRPEC strains (Fig.S5), which resulted in the reduction of bacterial virulence and fitness (Fig.6 and Fig. S7-8). Furthermore in response to this reviewer's comments we have now transformed wild-type *mcr-1* positive plasmids into wild-type non-MCRPEC human clinical strains (Chinese and European strains, Table S1) and showed that their virulence in a *G. mellonella* infection model is significantly reduced (Fig.7, see lines 297-307).

6) It has been shown in *A. baumannii* and *K. pneumoniae* that colistin-resistant strains can evolve to retain their virulence (Wand et al. 2015, Durante-Mangoni et al. 2015, Jones et al. 2017, and Kidd et al. 2017). While the colistin-resistance is not *mcr-1* mediated, but *pmrAB* or *mgrB* mediated instead, these papers should at least be cited in the discussion to show it is possible to have compensatory mutation in other species that retain, evolve to parent strain levels of virulence, or even lead to hyper-virulent strains (Kidd et al. 2017). These papers highlight how clinical isolates can evolve to accommodate LPS modification, and it is also in contrast to what is being presented in this submission. Meaning, compensatory mutants will likely evolve in the wild and likely have already evolved despite any negative impact of *mcr-1* expression on bacterial growth, which makes the results and conclusions being drawn by this manuscript less important. Therefore, not sure the data provides "hope" or not with regard to the clinical situation (Lines 399-407). Also, conveying "hope" seems incorrect when people have died when infected with strains containing *mcr-1*. Apparently, the clinical isolates are virulent enough to cause disease and mortality in patients so the conclusion being drawn from this work are out of place.

Reply: We agree with the reviewers suggestions and have added the recommended reference in lines 398-402. In comparison with wild-type MCRPEC parental strains, most of HLCRMs, except strains PN25 and PN42, showed no genomic changes in the following genes: *pmrE*, *pmrAB*, *pmrC*, *phoPQ*, *mgrB*, *arnB/D*, *cptA*, *eptB*, *ipxM*, *micA* and *arcAB*, which are the known to be associated with colistin resistance. However, our sequencing data from strains PN25 and PN42, have shown amino acid mutations that have been identified in PmrA (R81S) and PmrB (V161M), respectively. It would appear that both these chromosomal mutations are responsible for the high resistance to colistin. We have also examined the other five HLCRMs and not observed any mutation in the standard colistin resistance "loci" normally expected. We have added these findings into the discussion (lines 418-422). We are currently sequencing these strains with PacBio and minion to affirm our MiSeq data but we believe that is beyond the scope of this article.

7) Finally, the authors should evaluate the membrane proteins, lipid content, and phosphoethanolamine addition via mass spec or other types of analysis to show that indeed these modifications are linked to *mcr-1* activity and not artificial with respect to overexpression or serial passage. Non-specific activity and membrane rearrangement could account for differences seen, and not be directly linked to *mcr-1* expression.

Reply: In order to prove that bacterial fitness and virulence loss are specifically linked to *mcr-1* activity, MCRPEC transformants (MCRPECT) were generated as described above and their virulence comparison has been conducted. In comparison with non-MCRPEC, their counterpart MCRPECT carrying *mcr-1* plasmids do not affect bacterial growth of the host strain and so their fitness cost is modest (Fig.S9); however, their virulence is markedly depleted (Fig.7).

It should also be noted that we have purified LPS from MCRPEC and non-MCRPEC and reproducibly shown that the MCR-1 modified LPS stimulates immune factors (IL-6 and TNF) less when compared to the WT LPS consistent with the rest of the data presented in the manuscript. Taken together, these findings further support that *mcr-1* expression is responsible to bacterial virulence loss. We are unclear what the reviewer means by “non-specific activity and membrane rearrangement”. As MCR-1 directly modifies the core part of lipid A it seems logical that it would have some influence on immune stimulation (which we have reproducibly shown) and affects in-vivo virulence (which we have also repeatedly shown).

Minor Concerns

1) In the Introduction (Lines 64-67), it is pretty clear that the “spread” of *mcr-1* is linked to the increased surveillance. Clinical labs are doing well as far as tracking something that would be a tremendous threat to patients and caregivers. Therefore, the wording should reflect this and not say it is “uncertain”. Instead, it should say something to the effect of “likely due to increased surveillance”.

Reply: In accordance with this reviewer’s wishes, this sentence has been rewritten in lines 62-63.

2) Line 76-81 – The impact of *mcr-1* with respect to clinical data is difficult to reconcile, and here the Introduction should reflect that. While mortality rates could fluctuate, the patient status is rarely taken into account. The more compromised a patient is (immunocompromised or one or more comorbidities), the more likely a patient could suffer mortality from bacterial infection. So while mortality rates could be higher, it is truly difficult to tie this to the virulence of any given strain as the patient’s condition can vary. This point should at least be mentioned.

Reply: We fully agree with this reviewer’s comments and have cited (ref.10 and 15) and discussed the few clinical studies in the introduction (lines 73-81) and we are persuaded that the current clinical data is insufficient to draw definitive conclusions (lines 81-82).

3) Line 113-115 – Instead of listing all of these genes, found in the strain, it is better to just say “numerous antibiotic resistance genes” and cite the table in the supplemental material.

Reply: In accordance with this reviewer’s wishes, this sentence has been rewritten in lines 113-114.

4) Line 134 – the units, hours (“h”) should have a space between it and number “8”.

Reply: this error has now been corrected in line 133.

5) Line 413-414 – LB should be referred to as lysogeny broth...not Luria-Bertani. Bertani himself said so (Ezraty et al. 2014).

Reply: this error has now been corrected in line 452.

6) Is the same scale being used for each TEM picture in Fig. 4? B-2 and C-2 look more zoomed in when compared to A-2.

Reply: Yes, they are the same scale.

7) Fig. 9A – *Mellonella* should not be capitalized, should be “mellonella”

Reply: this error was corrected in fig.8A

Reviewer #2 (Remarks to the Author):

In this manuscript Yang et al. provide data to support their claim that the expression of *mcr-1* in *E. coli* results in reduced bacterial fitness, and the authors thus hypothesize that the expression of *mcr-1* must be tightly regulated in order to strike an equilibrium between maintaining colistin resistance and bacterial fitness. Specifically, the authors provide evidence that *mcr-1* expression and over-expression reduces bacterial growth rate, viability and fitness in direct competition experiments. They also provide evidence that *mcr-1* expression results in changes in bacterial cell morphology and that the modifications in lipopolysaccharide that occur due to *mcr-1* expression result in LPS that has less ability to stimulate cytokine release from macrophages. In addition, the authors provide evidence that strains containing the *mcr-1* gene, which typically demonstrate low-level colistin resistance (MICs 2-8), can develop high level colistin resistance (MICs > 64) after selection in the presence of increasing concentrations of colistin. Given the increasing clinical significance of colistin resistance, especially due to *mcr-1* expression, these data are of potential interest to others in the field. In the absence of previous studies performing a detailed analysis of the fitness cost of *mcr-1* expression, these data represent an important degree of novelty. However, there are some methodological details which raise doubts regarding the conclusions drawn by the authors and should be clarified (detailed in Major Comments).

Major Comments.

1. In Figure 3A the authors assess growth effects caused by increasing expression of *mcr-1*. However, the authors begin the experiments with an inoculum of $9.6 \log_{10}$ CFU of bacteria which would appear to provide little margin for measuring growth during the exponential phase. Can the authors justify using such a high initial concentration?

Reply: We induced *mcr-1* expression at different phases (lag phase and log phase) of bacteria. The results showed no difference whether *mcr-1* was induced in either the lag phase or the log phase. In keeping with this reviewer's concerns, we have now added a new growth curve with *mcr-1* induced at lag phase to Fig.2A.

2. In Figures 3B and 3C data from control experiments performed in the same way with the strain carrying an empty plasmid (TOP10/pBAD) in order to rule out effects caused by increasing concentrations of arabinose.

Reply: We have used *E. coli* TOP10/pBAD induced with 0.2% arabinose and showed no effects on either cell membrane (Fig.S3) and cell viability (Fig.4). Please also refer to our detailed response to reviewer 1.

3. Do the authors have any evidence that continuing to increase the expression of *mcr-1* beyond a certain point results in additional modification of LPS (for example, does increasing the expression of *mcr-1* from 358-fold to 1132-fold, for example in Figure 3B, increase the addition of ethanolamine to LPS)? If not, the growth and fitness defects observed upon additional expression may simply be due to the additional energy expenditure necessary for the overexpression of a protein.

Reply: The reviewer raises a very good point. We believe that the addition of ethanolamine to the LPS has a fitness and virulence cost as witnessed by the evidence produced in our article. However, the amount that can be modified and tolerated by the *E. coli*, remains unknown - we are currently working on this aspect and have some MALDI-TOFF data on crude LPS extracts which is qualitative but not quantitative. We believe there is a saturation rate for LPS modification. The changes seen in fitness etc. cannot simply be down to energy expenditure as the TEM (Fig. 3C) show gross changes in the LPS which will have a profound effect on normal cell physiology.

4. In Figure 4 the authors should provide images from the strain containing the empty plasmid (TOP10/pBAD) in the presence of arabinose in order to rule out any morphological changes due to high arabinose concentrations.

Reply: We have added the new figure. S3, *E. coli* TOP10/pBAD alone induced with 0.2% arabinose and showed no effects on cell membrane (lines 166-170). Please see also our detailed reply to reviewer 1.

5. In the data presented in Figure 6 it is unclear why different graphs are used for presenting data for IL-6. Can the authors provide a justification. Additionally, why are the same time points and LPS concentrations that were used for measuring IL-6 concentrations not shown for TNF-alpha concentrations.

Reply: We fully accept this criticism; however, because the cells have been stimulated by LPS at lower concentrations (0.045 and 0.0045 ng/ml), no difference could be shown in the production of TNF-alpha. We therefore have adjusted these figures to accommodate the sensitivity of the assay.

6. A major question regarding the data presented in this manuscript is the reason for the increased colistin resistance seen in strains containing *mcr-1*. After selection of these strains in the presence of

colistin and observing dramatic increases in MIC values, the authors state that they detect no changes in the expression of *mcr-1* (except in one case), and no mutations in *mcr-1* or surrounding sequences. Could there be mutations in other determinants of colistin resistance (e.g. *pmrA/pmrB*). What is the explanation for the phenotypic changes in these strains?

Reply: In accordance with this reviewer's wishes, we added this information in line 417-423. When compared to their respective wild-type parental strains, most of HLCRMs, except strains PN25 and PN42, showed no genomic changes in the following genes: *pmrE*, *pmrAB*, *pmrC*, *phoPQ*, *mgrB*, *arnB/D*, *cptA*, *eptB*, *ipxM*, *micA* and *arcAB*, which are known to be associated with colistin resistance. However, in strains PN25 and PN42, amino acid mutations have been identified in PmrA (R81S) and PmrB (V161M), respectively.

7. The authors use the *Galleria mellonella* model to assess decreases in pathogenicity of the *mcr-1* expressing strains. Although this model has been validated for assessing fitness of bacterial strains, is it the most appropriate model given the fact that the authors claim that one of the main differences between strains expressing *mcr-1* and those without *mcr-1* is the ability to stimulate the immune response?

Reply: The use of the *Galleria* model allows us to test large numbers of larvae in a single time point and therefore is a more statistically sound model than a murine model. It is also more ethically acceptable. However, the reviewer has a valid point and to address his/her concerns we would argue that we have examined ethanolamine modified LPS and native LPS and showed the MCR-1 modified LPS has less immune stimulation (IL-6 and TNF) (Fig.5).

Furthermore, In order to offer more compelling evidence for the impact of MCR-1 expression on the virulence and fitness, we have transferred *mcr-1* positive plasmids into non-MCRPEC human clinical strains. Our data shows that the new *mcr-1*-positive transformants (MCRPECT) confers not only slow growth (Fig.S9), but also statistically significant reduced bacterial virulence by increasing the survival rate in a *G. mellonella* infection model (Fig.7).

Minor Comments.

1. The title is somewhat confusing as it is not clear what "essential cellular defence mechanisms" refers to.

Reply: We chose the title to appeal to the general readership and to articulate the balance the cell has to choose between defending itself from the effects of colistin yet ensuring that the expression of the gene is suitably controlled as to minimalise its toxic effects. We are open to alternative suggestions but would still wish the title to appeal to a general audience.

2. There are a number of figures which may not be necessary for supporting the author's claims, for examples, Figure 1B, Figure 2 B and C, and Supplementary Figure 1.

Reply: we have moved fig.2 to the supplementary figure.S2. But we believe that Figure 1B and Figure S1 should stay in the main text and Supplementary materials, respectively. The Figure S1 graphically

demonstrates the global dissemination of *mcr-1* gene, and Figure 1B can help us to picture the molecular mechanism of colistin binding activity (left) and *mcr-1* mediated colistin resistance (right).

3. In the discussion the authors claim that their data suggest that increased leakage of cellular cytoplasm occurs upon expression of *mcr-1* (line 397). It is unclear how the data presented in Figure 4 support this conclusion.

Reply: We have drawn these conclusions from the data represented in Figure 4C (now is the Fig.3C1), the electron density (and therefore cellular integrity) in the cell cytoplasm is completely lost, compared with the two negative controls (fig.3A1-B1), and the gross cell membrane changes can be observed in figure 4C2 where the membrane integrity is completely destroyed. .

4. The manuscript contains a number of typographical errors and would benefit from improving the level of English that is used.

Reply: Typographic corrections have been made through the whole manuscript although Reviewer 1 did comment that the paper was “well written”.

Summary

In the newly revised manuscript by Yang et al., the authors continue to define the impact on virulence that *mcr-1* brings to bear on bacteria, specifically a variety of *Escherichia coli* strains subjected to *mcr-1* expression off of clinical plasmids or pBAD overexpression. Improvements to the manuscript now include the use of clinical isolates in different genomic contexts and genome sequencing to identify other mutations. In particular, the data found in Fig. 7, Fig. S9, and Fig. S5 show the impact of native *mcr-1* plasmids on virulence, and the authors should be commended for this inclusion. The addition of this experimental data improves the article. The authors also did a great job of reorganizing the manuscript, moving figures to the Supplemental Data, which helps to keep the key data upfront and provide a better overall flow for the reader. However, there are still major and minor concerns regarding this study.

Major concerns

- 1) There is still one major concern regarding this work: the use of pBAD expression to study the impact of *mcr-1* on fitness/virulence. I will try to make this as clear as possible. Despite the authors reply, it is almost certainly true that any gene overexpressed with pBAD is known to impact bacterial growth (and consequently virulence). This is true of even innocuous, soluble protein products such as GFP and RFP (Ghodke et al., 2016; Sigele and Hu, 1997). The authors cite beta-lactamase expression in their reply, but those proteins are often secreted and do not necessarily “gum up” the internal works of the bacterium. Importantly, MCR-1 has also been shown by Sun et al. 2017 to be a membrane protein found in the periplasmic space. Proteins with transmembrane domains are most certainly toxic (especially if the gene is being expressed without a cognate chaperone or other targeting/assemblies). Take this from someone who has also used pBAD to express proteins with transmembrane domains for 17+ years to complement knockout mutations. I can cite countless personal examples, but that negative data is not published and methods to address the issue were always implemented. Many others that have used pBAD for similar genetics experiments could also cite issues where proteins aggregated and too much of the target protein can be toxic to the cell. Perhaps the best publication regarding this is Baneyx, 1999.

With this in mind, it is likely that the toxicity, especially at saturating concentrations (0.2%) of expression by pBAD (Széliová et al. 2016) is solely mediated by the bacterium’s protease/degradation machinery (Baneyx 1999) not being able to keep up with the expression and erroneous localization of MCR-1 in the cytoplasm. Often, bacteria attempt to counter this overexpression by targeting toxic proteins to inclusion bodies. In this case, the end result being observed (Fig. 2 – Fig. 6) could be cell death or delays in replication, not mediated by enzymatic activity of MCR-1 or LPS modification (as the authors

would have us believe), rather an artifact from overexpression of a membrane protein.

Based on this, the burden is on the authors to explain how the fitness cost/virulence defects being observed from *mcr-1* being expressed off of pBAD at an arabinose concentration beyond 0.001% is nothing more than an artifact. Bayer et al. 1995 showed that maximal induction occurred at 0.001% arabinose. So again, to be clear and summarize how this reviewer sees the results in Fig. 2- Fig. 6:

Arabinose Concentration (%)	Type of expression (low, medium, high, saturating)	Toxic (mild/medium/high)	Reason for toxicity
0	Zero - Low	No	n/a
0.0002	Low - Medium	Mild	LPS modification?
0.002	Medium - Maximal	Medium	LPS modification?
0.02	High	High	Overexpression Artifact
0.2	Saturating	High	Overexpression Artifact

To further have the authors understand the possible artifact of the system being used, the pBAD vector was cited to be purchased from Thermofisher (Line 462). So rather than using a low-copy number pBAD vector such as pBAD33 with a pACYC backbone with 30-50 copies per cell (Guzman et al., 1995), this commercial version uses a pUC origin of replication with at least 1000 copies per cell. How does this make Fig. 2- Fig. 6 physiologically relevant?

The authors themselves in the Introduction say that *E. coli* only maintain *mcr-1* on low copy plasmids (copy number 1-5) and cite one strain with just one copy on the chromosome. If that is the case, why should we care about this artificial pBAD expression at all? Fig. 7 at least shows an effect on virulence not artificially driven.

In order to address this concern, the authors need to address two things: (a) consider another pBAD plasmid with a lesser copy number. (b) reduce the amount of arabinose being used to a more nominal amount. Perhaps (b) is the easier fix for this manuscript, but all of the data with 0.2% and 0.02 arabinose would have to be redone at a lower concentration (0.001 or 0.002%) as a true comparator or just removed. Points 2 and 3 below will also help. At a minimum, the authors should at least acknowledge that overexpression at saturating concentrations could generated an artificial killing of the cell.

- 2) Also, the authors have still disregarded the need to construct a proper negative control for pBAD::*mcr-1*. The authors have not cloned another gene into pBAD (housekeeping, GFP, periplasmic membrane, beta-lactamase as they suggest, etc.) to generate a comparator strain that could be used to compare arabinose/expression levels to evaluate fitness/virulence side-by-side in each of the assays. The best comparator would be another enzyme that also has transmembrane proteins that targets the periplasm. Without this control, it is possible that overexpression of *mcr-1* leads to improper folding, protein aggregation, mislocalization, etc. that leads to cell death as mentioned above. These outcomes have nothing to do with the enzyme's proper localization and function and call in to question the major conclusion the authors are making. If controlled properly, there would be less of an issue regarding this data.
- 3) Finally, the authors present no way to confirm the phosphoethanolamine addition is occurring to the LPS after induction with pBAD or with clinical isolate transformation. While it is fine to assume the addition is happening given the presence of the enzyme and deleterious phenotypes, doing the MALDI-TOF to confirm the LPS modification would strengthen the manuscript. Also, the authors mention there is an "all or none" phenotype in the response to reviewers. Again, this is exactly what is seen with pBAD (Siegele and Hu, 1997). Some individual bacteria in the population express the protein, and others do not (Morgan-Kiss et al., 2002). This fits with their data, but it further stresses the need to find out if the LPS modification/phosphoethanolamine addition is happening at a lower concentration of arabinose (0.001) to avoid other artifacts that complicate their findings at higher concentrations (>0.02%) and call into question the reason for cell death.

Minor Concerns

- 1) There is no doubt the contribution of *Galleria mellonella* to studying virulence is tremendously important for a number of reasons (ethical being primary), but it can never fully replace experiments in mammalian systems as there are significant differences in the mammalian system that contribute to virulence. In this case, maybe it doesn't apply as much since the innate immune system in both animals and have a somewhat similar pathway (TLR4), but it is still better to confirm. In short, what happens in *G. mellonella* does not always happen in mice, and vice-versa. So the editor may have granted a reprieve, but it doesn't change the fact this is still a significant weakness of this work. Not to mention, because *G. mellonella* are cheap and easy to work with, most people often do 20-25 worms/group. Ten animals/group provides appropriate statistical power here, but the authors could have used more worms/group.
- 2) In the abstract, it would be better to use the full name *Galleria mellonella*, and italicize it properly. The full name should also be used in the Introduction, and from then on, *G. mellonella* can be utilized. Please check throughout the manuscript to make sure this is consistent.

- 3) Line 99-100 – should be “its ability”. Please review English again at various places. There were some other errors such as this.
- 4) Line 116-117 – There should be a space after each period, even for “Fig. S2”.
- 5) Line 117 – 120 – The sentence would be better served as “The MLST for each MCRPEC strain were grouped as follows: ST2040 (PN16)...”
- 6) Line 133-135 – The sentence should be “Maximal *mcr-1* induction was observed at 0.02% L-arabinose, where *mcr-1* expression was approx. 2-fold and 3-fold more than the other inducing concentrations of 0.002% and 0.2%, respectively.”
- 7) Line 136-138 – Please use \log_{10} , and also the inoculum was what? 9.6×10^7 CFU? I assume it is 9.6×10^5 , and this is a typo. The other reviewer brought this up. If it is 9.6×10^{10} . This is huge starting amount and calls into question the method.
- 8) Figure 5. “The concentrations of IL-6 produced by macrophages induced by unmodified LPS were consistently higher than IL-6 levels produced by MCR-1 modified LPS at 8 and 24h (Fig. 5A and B). Additionally, TNF-alpha levels were also higher in macrophages stimulated by unmodified LPS than compared to modified LPS (Fig. 5C).”

This seems to be only seen with one single data point. The SD bars overlap. Statistics are not done correctly, which means this conclusion cannot be made.

- 9) Line 459-460 - Still have it as “Lysogeny-Bertani”. Should just be “lysogeny broth (LB)”

Reviewer #2 (Remarks to the Author):

The manuscript is greatly improved with the additional information and data provided by the authors, especially with respect to the additional control data that was requested. There are still a couple of important outstanding questions:

1. What is the genetic basis for the dramatic change in colistin susceptibility seen in the *mcr-1* containing strains that were selected for in the presence of colistin? (Major comment 6)
2. Is electron microscopy sufficient for drawing the conclusion that increased cellular leakage is occurring upon expression of *mcr-1*? (Minor comment 3)

However, the authors have adequately addressed these questions and limitations with the additional data and explanation provided in the text.

Summary

In the newly revised manuscript by Yang et al., the authors continue to define the impact on virulence that *mcr-1* brings to bear on bacteria, specifically a variety of *Escherichia coli* strains subjected to *mcr-1* expression off of clinical plasmids or pBAD overexpression. Improvements to the manuscript now include the use of clinical isolates in different genomic contexts and genome sequencing to identify other mutations. In particular, the data found in Fig. 7, Fig. S9, and Fig. S5 show the impact of native *mcr-1* plasmids on virulence, and the authors should be commended for this inclusion. The addition of this experimental data improves the article. The authors also did a great job of reorganizing the manuscript, moving figures to the Supplemental Data, which helps to keep the key data upfront and provide a better overall flow for the reader. However, there are still major and minor concerns regarding this study.

Major concerns

- 1) There is still one major concern regarding this work: the use of pBAD expression to study the impact of *mcr-1* on fitness/virulence. I will try to make this as clear as possible. Despite the authors reply, it is almost certainly true that any gene overexpressed with pBAD is known to impact bacterial growth (and consequently virulence). This is true of even innocuous, soluble protein products such as GFP and RFP (Ghodke et al., 2016; Sigele and Hu, 1997). The authors cite beta-lactamase expression in their reply, but those proteins are often secreted and do not necessarily “gum up” the internal works of the bacterium. Importantly, MCR-1 has also been shown by Sun et al. 2017 to be a membrane protein found in the periplasmic space. Proteins with transmembrane domains are most certainly toxic (especially if the gene is being expressed without a cognate chaperone or other targeting/assemblies). Take this from someone who has also used pBAD to express proteins with transmembrane domains for 17+ years to complement knockout mutations. I can cite countless personal examples, but that negative data is not published and methods to address the issue were always implemented. Many others that have used pBAD for similar genetics experiments could also cite issues where proteins aggregated and too much of the target protein can be toxic to the cell. Perhaps the best publication regarding this is Baneyx, 1999.

With this in mind, it is likely that the toxicity, especially at saturating concentrations (0.2%) of expression by pBAD (Széliová et al. 2016) is solely mediated by the bacterium’s protease/degradation machinery (Baneyx 1999) not being able to keep up with the expression and erroneous localization of MCR-1 in the cytoplasm. Often, bacteria attempt to counter this overexpression by targeting toxic proteins to inclusion bodies. In this case, the end result being observed (Fig. 2 – Fig. 6) could be cell death or delays in replication, not mediated by enzymatic activity of MCR-1 or LPS modification (as the authors

would have us believe), rather an artifact from overexpression of a membrane protein.

Based on this, the burden is on the authors to explain how the fitness cost/virulence defects being observed from *mcr-1* being expressed off of pBAD at an arabinose concentration beyond 0.001% is nothing more than an artifact. Bayer et al. 1995 showed that maximal induction occurred at 0.001% arabinose. So again, to be clear and summarize how this reviewer sees the results in Fig. 2- Fig. 6:

Arabinose Concentration (%)	Type of expression (low, medium, high, saturating)	Toxic (mild/medium/high)	Reason for toxicity
0	Zero - Low	No	n/a
0.0002	Low - Medium	Mild	LPS modification?
0.002	Medium - Maximal	Medium	LPS modification?
0.02	High	High	Overexpression Artifact
0.2	Saturating	High	Overexpression Artifact

To further have the authors understand the possible artifact of the system being used, the pBAD vector was cited to be purchased from Thermofisher (Line 462). So rather than using a low-copy number pBAD vector such as pBAD33 with a pACYC backbone with 30-50 copies per cell (Guzman et al., 1995), this commercial version uses a pUC origin of replication with at least 1000 copies per cell. How does this make Fig. 2- Fig. 6 physiologically relevant?

The authors themselves in the Introduction say that *E. coli* only maintain *mcr-1* on low copy plasmids (copy number 1-5) and cite one strain with just one copy on the chromosome. If that is the case, why should we care about this artificial pBAD expression at all? Fig. 7 at least shows an effect on virulence not artificially driven.

In order to address this concern, the authors need to address two things: (a) consider another pBAD plasmid with a lesser copy number. (b) reduce the amount of arabinose being used to a more nominal amount. Perhaps (b) is the easier fix for this manuscript, but all of the data with 0.2% and 0.02 arabinose would have to be redone at a lower concentration (0.001 or 0.002%) as a true comparator or just removed. Points 2 and 3 below will also help. At a minimum, the authors should at least acknowledge that overexpression at saturating concentrations could generated an artificial killing of the cell.

Reply. pBAD expression systems and MCR-1 controls. Similar to reviewer #1, the senior author too has been working on pBAD expression systems from his post-doc days in London in the mid-1990's and have overexpressed over 30 different genes/mutants for the purpose of protein purification. Needless to say, we do not share this reviewer's concerns on the copy number or leakage of pBAD promoter and the burden of over-expressing any protein. The pBAD-HisA plasmids (Invitrogen, UK) used in our study contains the pBR322 origin pMB1, which is making it a low copy number (15-20 copies/cell). This information is available online (<https://www.thermofisher.com/order/catalog/product/V43001>). All our evidence suggests that the leakage of pBAD is minimal to non-existent. To counter reviewer 1 comment's "it is almost certainly true that any gene overexpressed with pBAD is known to impact bacterial growth (and consequently virulence", and in addition to the suggestions of the editor, we have over-expressed (pBAD with 0.2% arabinose) *bla*_{TEM-1} (the most common β -lactamase worldwide) and repeated all experiments pertaining to growth/fitness/toxicity etc. As can be seen from our data (Fig.5c, S4c and S5c), the overexpression of *bla*_{TEM-1} has little or zero impact on the fitness (toxic effect on bacterial growth, cell viability and morphological changes) of the *E. coli* strain - as predicted. Accordingly, all effects seen with MCR-1 are due to: 1. The embedding of the MCR-1 structural protein into the *E. coli* outer-membrane and/or 2. The enzymatic activity of MCR-1 in modifying the core of lipid A. We also disagree with the reviewers final comments on the use of pBAD as we find this an extremely useful system to control and monitor the effects of high-expression of a single gene.

As suggested by the editor, we have over-expressed the E246A knock-out mutant (coordinating the zinc ion) and repeated growth/fitness/toxicity with this construct. Additionally, we also repeated the same series of experiments with the MCR-1 soluble fraction to address whether it is MCR-1 in its entirety or its trans-membrane spanning region that causes the toxic effects we observe - this distinction is important. We provide new data (Fig.5, S4 and S5) to confirm that the *mcr-1* mediated fitness cost is due to both the embedding of the protein and the modification of LPS although we cannot ascribe the direct proportion of these activities.

MCR-1 is a membrane-bound enzyme consisting of five hydrophobic transmembrane helices and a soluble periplasmic domain. To examine whether the transmembrane domain or MCR-1 catalytic domain play a role on bacterial fitness, the mutation implicated in MCR-1 active site, E246A, and MCR-1 soluble domain (residues 219-541 - lacking the N-terminal membrane-bound region), were cloned into pBAD-hisA plasmid. Results show that the MCR-1 E246A mutant caused some cellular toxic effect as observed by a decreased growth rate (Fig.S4), fitness loss (Fig.7) and reduced cell viability (Fig.5). However, the overexpression of MCR-1 soluble domain did not show any slow growth, fitness loss or toxic effect on bacteria (Fig. 5 and Fig. S-). The toxicity of MCR-1 (E246A) mutant is moderate, compared to the impact of the fully active MCR-1 protein (Fig. 2-4). For example, wild-type MCR-1 caused approx. 0.33 bacterial death, in contrast, less than 0.09 of cells have been killed by MCR-1 (E246A) mutant (Fig. 5). Therefore, our data shows that the fitness loss and bacterial toxicity is due to both the embedding of the protein and the modification of LPS, which is mediated by catalytic domain of MCR-1 enzyme.

2) Also, the authors have still disregarded the need to construct a proper negative

control for pBAD::*mcr-1*. The authors have not cloned another gene into pBAD (housekeeping, GFP, periplasmic membrane, beta-lactamase as they suggest, etc.) to generate a comparator strain that could be used to compare arabinose/expression levels to evaluate fitness/virulence side-by-side in each of the assays. The best comparator would be another enzyme that also has transmembrane proteins that targets the periplasm. Without this control, it is possible that overexpression of *mcr-1* leads to improper folding, protein aggregation, mislocalization, etc. that leads to cell death as mentioned above. These outcomes have nothing to do with the enzyme's proper localization and function and call in to question the major conclusion the authors are making. If controlled properly, there would be less of an issue regarding this data.

Reply. We have construct three different pBAD plasmids to address this issue. Firstly, one containing *bla*_{TEM-1} and overexpressed by 0.2% arabinose to evaluate its impact on fitness and bacterial viability. The results unequivocally show (Fig. 5 and S4) that the expression of MCR-1 is responsible for the bacterial toxicity, rather than the burden of protein expression *per sae* as opinionated by this reviewer.

Secondly, we sub-cloned a *mcr-1*-inactive mutant (E246A, Hinchliffe, P. et al 2017, Sci report) into pBAD as suggested by the editor. Despite the fact that this mutant lacks any activity (null zinc content), its toxic effect still can be observed by growth arrest (Figure S4) and cell viability (figure 5), which suggests that the toxicity of MCR-1 protein may be, in part, attributable to its transmembrane domain. To confirm this hypothesis, we cloned only soluble domain of *mcr-1*(encoding residues 219-541) into pBAD. The results (figure 5, S4 and S5) indicate that the bacteria behave as the negative controls with good growth rate and cellular morphology.

- 3) Finally, the authors present no way to confirm the phosphoethanolamine addition is occurring to the LPS after induction with pBAD or with clinical isolate transformation. While it is fine to assume the addition is happening given the presence of the enzyme and deleterious phenotypes, doing the MALDI-TOF to confirm the LPS modification would strengthen the manuscript. Also, the authors mention there is an "all or none" phenotype in the response to reviewers. Again, this is exactly what is seen with pBAD (Siegele and Hu, 1997). Some individual bacteria in the population express the protein, and others do not (Morgan-Kiss et al., 2002). This fits with their data, but it further stresses the need to find out if the LPS modification/phosphoethanolamine addition is happening at a lower concentration of arabinose (0.001) to avoid other artifacts that complicate their findings at higher concentrations (>0.02%) and call into question the reason for cell death.

Reply. We have provided new evidence (Fig. S12) to confirm the phosphoethanolamine (PEA) addition onto lipidA by ESI-MS/MS, as suggested by the editor. In Fig. S12, ESI-MS/MS spectrum visualization of the negative ion of the lipid A extracted from the constructed *mcr-1*positive strains. A PEA (123u) is added into the bis-phosphorylated

hexaacylated lipid A (m/z 1920).

Minor Concerns

- 1) There is no doubt the contribution of *Galleria mellonella* to studying virulence is tremendously important for a number of reasons (ethical being primary), but it can never fully replace experiments in mammalian systems as there are significant differences in the mammalian system that contribute to virulence. In this case, maybe it doesn't apply as much since the innate immune system in both animals and have a somewhat similar pathway (TLR4), but it is still better to confirm. In short, what happens in *G. mellonella* does not always happen in mice, and vice-versa. So the editor may have granted a reprieve, but it doesn't change the fact this is still a significant weakness of this work. Not to mention, because *G. mellonella* are cheap and easy to work with, most people often do 20- 25 worms/group. Ten animals/group provides appropriate statistical power here, but the authors could have used more worms/group.

Reply: here we can provide a lot of evidence indicating that there is a good correlation between virulence study in the insect host and the virulence measured through systemic infection of mice (such as ref. M Brennan et al. 2002; Kimberley D. Seed et al 2008).

- 2) In the abstract, it would be better to use the full name *Galleria mellonella*, and italicize it properly. The full name should also be used in the Introduction, and from then on, *G. mellonella* can be utilized. Please check throughout the manuscript to make sure this is consistent.

Reply: all typos (highlighted in red) have been corrected throughout the manuscript.

- 3) Line 99-100 – should be “its ability”. Please review English again at various places. There were some other errors such as this.

Reply: all minor typos (highlighted in red) have been corrected throughout the manuscript.

- 4) Line 116-117 – There should be a space after each period, even for “Fig. S2”.

Reply: all minor typos (highlighted in red) have been corrected throughout the manuscript.

- 5) Line 117 – 120 – The sentence would be better served as “The MLST for each MCRPEC strain were grouped as follows: ST2040 (PN16)....”

Reply: the sentence has been written as suggestion, line117-118.

- 6) Line 133-135 – The sentence should be “Maximal *mcr-1* induction was observed at 0.02% L-arabinose, where *mcr-1* expression was approx. 2-fold and 3-fold more than the other inducing concentrations of 0.002% and 0.2%, respectively.”

Reply: the sentence has been written as suggestion, line134-135.

- 7) Line 136-138 – Please use log₁₀ , and also the inoculum was what? 9.6×10^7 CFU? I assume it is 9.6×10^5 , and this is a typo. The other reviewer brought this up. If it is 9.6×10^{10} . This is huge starting amount and calls into question the method.

Reply: we have repeated the growth curve (fig 2A), induced *mcr-1* expression with 7.5 log₁₀ (CFU/ml) initial inoculum

- 8) Figure 5. “The concentrations of IL-6 produced by macrophages induced by unmodified LPS were consistently higher than IL-6 levels produced by MCR-1 modified LPS at 8 and 24h (Fig. 5A and B). Additionally, TNF-alpha levels were also higher in macrophages stimulated by unmodified LPS than compared to modified LPS (Fig. 5C).”

This seems to be only seen with one single data point. The SD bars overlap. Statistics are not done correctly, which means this conclusion cannot be made.

Reply: IL-6 and TNF-alpha assay have been repeated triplicate and duplicate, error bars can be observed clearly in Fig 5A-C. as suggested.

- 9) Line 459-460 - Still have it as “Lysogeny-Bertani”. Should just be “lysogeny broth (LB)”

Reply: This typo (highlighted in red) has been corrected.

REVIEWERS' COMMENTS:

Reviewer #1 (Remarks to the Author):

In the newest iteration of the manuscript by Yang et al., the authors met many of the previous reviewers' requests. In fact, at this time, there are just minor concerns and few points of issue. Kudos to the authors for doing the work required to the manuscript to this current state. The controls are better, and the inclusion of mass spec to determine the phosphoethanolamine addition truly strengthened the manuscript. See below:

- 1) The very first sentence of the Abstract should probably be "a lipid A modifying enzyme" not "mechanism".
- 2) Line 97 – there should be a space between "lipid" and "a".
- 3) Line 211 – should be TOP10, all capitalized.
- 4) Line 394-399 - this is a really important point and nice to see its inclusion.
- 5) Line 397 – there are two periods at the end of the sentence.
- 6) Line 425 – Salmonella should be italicized.
- 7) Line 683 – there should be a space after "1" unit.
- 8) Line 714 – there should be a space between "x" and "g"
- 9) Line 716 – there should be a space between pH and 4.5
- 10) Line 692 – should be mL not ml and a space after the number "2". Same idea, Line 719 should be μL . Not μl . Which is also the case with microliters and milliliters at lines: 675, 668, 669, 645, 630, 598, 587, 559, 560, 520, 510, 294. In fact, the one at Line 254 and 547 may be the only one that is correct. Please check all of these.

Point-by-point response to reviewers' comments:

Reviewer #1 (Remarks to the Author):

In the newest iteration of the manuscript by Yang et al., the authors met many of the previous reviewers' requests. In fact, at this time, there are just minor concerns and few points of issue. Kudos to the authors for doing the work required to the manuscript to this current state. The controls are better, and the inclusion of mass spec to determine the phosphoethanolamine addition truly strengthened the manuscript. See below:

1) The very first sentence of the Abstract should probably be “a lipid A modifying enzyme” not “mechanism”.

Reply: it has been corrected in line 34

2) Line 97 – there should be a space between “lipid” and “a”.

Reply: corrections of this kind of typos have been made throughout the manuscript.

3) Line 211 – should be TOP10, all capitalized.

Reply: corrected in line 219

4) Line 394-399 - this is a really important point and nice to see its inclusion.

Reply: Thank you

5) Line 397 – there are two periods at the end of the sentence.

Reply: corrected in line 417.

6) Line 425 – Salmonella should be italicized.

Reply: corrected in line 439.

7) Line 683 – there should be a space after “1” unit.

8) Line 714 – there should be a space between “x” and “g”

9) Line 716 – there should be a space between pH and 4.5

10) Line 692 – should be mL not ml and a space after the number “2”. Same idea, Line 719 should be μL . Not μl . Which is also the case with microliters and milliliters at lines: 675, 668, 669, 645, 630, 598, 587, 559, 560, 520, 510, 294. In fact, the one at Line 254 and 547 may be the only one that is correct. Please check all of these.

Reply: this kind of corrections (comments 7-10) has been made throughout the manuscript.